# Asymptotic Guarantees for Generative Modeling Based on the Smooth Wasserstein Distance

**Ziv Goldfeld**
Cornell University
goldfeld@cornell.edu

**Kristjan Greenewald**
MIT-IBM Watson AI Lab
kristjan.h.greenewald@ibm.com

**Kengo Kato**
Cornell University
kk976@cornell.edu

## Abstract

Minimum distance estimation (MDE) gained recent attention as a formulation of (implicit) generative modeling. It considers minimizing, over model parameters, a statistical distance between the empirical data distribution and the model. This formulation lends itself well to theoretical analysis, but typical results are hindered by the curse of dimensionality. To overcome this and devise a scalable finite-sample statistical MDE theory, we adopt the framework of smooth 1-Wasserstein distance (SWD) $\mathsf{W}_1^{(\sigma)}$. The SWD was recently shown to preserve the metric and topological structure of classic Wasserstein distances, while enjoying dimension-free empirical convergence rates. In this work, we conduct a thorough statistical study of the minimum smooth Wasserstein estimators (MSWEs), first proving the estimator's measurability and asymptotic consistency. We then characterize the limit distribution of the optimal model parameters and their associated minimal SWD. These results imply an $O(n^{-1/2})$ generalization bound for generative modeling based on MSWE, which holds in arbitrary dimension. Our main technical tool is a novel high-dimensional limit distribution result for empirical $\mathsf{W}_1^{(\sigma)}$. The characterization of a nondegenerate limit stands in sharp contrast with the classic empirical 1-Wasserstein distance, for which a similar result is known only in the one-dimensional case. The validity of our theory is supported by empirical results, posing the SWD as a potent tool for learning and inference in high dimensions.

## 1 Introduction

Minimum distance estimation (MDE) considers the minimization of a statistical distance (SD) between the empirical data distribution and a parametric model class. Given an identically and independently distributed (i.i.d.) dataset $X_1, \ldots, X_n$ sampled from $P$, the goal is to learn a model $Q_\theta$, for $\theta \in \Theta$, that approximates the empirical measure $P_n := n^{-1} \sum_{i=1}^n \delta_{X_i}$ under a SD[1] $\delta$, i.e., we aim to find $\widehat{\theta}_n \in \operatorname{argmin}_{\theta \in \Theta} \delta(P_n, Q_\theta)$. This classic mathematical statistics problem [1–3] was adopted in recent years as a formulation of generative modeling. Indeed, both generative adversarial networks (GANs) [4–11] and variational (or Wasserstein) autoencoders [12, 13] stem from different strategies for (approximately) solving MDE[2] for various choices of $\delta$.

Beyond the practical effectiveness of MDE-based generative models, this formulation is well-suited for a theoretic analysis. This inspired a recent line of works studying GAN generalization in terms of MDEs [9, 14, 15]. Such sample-complexity results boil down to the rate of empirical approximation under the chosen SD, i.e., the speed at which $\delta(P_n, P)$ converges to zero. Unfortunately, popular SDs, such as Wasserstein distances [16], $f$-divergences [17], and integral probability metrics [18]

(excluding maximum mean discrepancy [19]) suffer from the curse of dimensionality (CoD), converging as $\delta(P_n, P) \asymp n^{-1/d}$, with $d$ being the data dimension [20–23].[3] This limits the practical usefulness of the devised results, which degrade exponentially fast with dimension.

## 1.1 MDE with Smooth Wasserstein Distance and Contributions

To circumvent the CoD impasse, we adopt the smooth 1-Wasserstein distance (SWD) [27,28] as our SD. Namely, for any $\sigma > 0$, consider $\mathsf{W}_1^{(\sigma)}(P, Q) := \mathsf{W}_1(P * \mathcal{N}_\sigma, Q * \mathcal{N}_\sigma)$, where $\mathcal{N}_\sigma = \mathcal{N}(0, \sigma^2 \mathrm{I}_d)$ is the $d$-dimensional isotropic Gaussian measure of parameter $\sigma$, $P * \mathcal{N}_\sigma$ is the convolution of $P$ and $\mathcal{N}_\sigma$, and $\mathsf{W}_1$ is the regular 1-Wasserstein distance (see Section 2 for details). The motivation for this choice is twofold. First, the 1-Wasserstein distance is widely used for generative modeling [8, 13, 29, 30] due to its beneficial attributes, such as metric structure, robustness to support mismatch, compatibility to gradient-based optimization, etc. As shown in [28], these properties are all preserved under Gaussian smoothing. Second, while $\mathsf{W}_1$ suffers from the CoD, [27] showed that $\mathbb{E}\big[\mathsf{W}_1^{(\sigma)}(P_n, P)\big] \lesssim_{\sigma,d} n^{-1/2}$ in all dimensions, whenever $P$ is sub-Gaussian.[4] The considered minimum smooth Wasserstein estimator (MSWE) is thus

$$\widehat{\theta}_n \in \operatorname*{argmin}_{\theta \in \Theta} \mathsf{W}_1^{(\sigma)}(P_n, Q_\theta). \tag{1}$$

We first prove measurability and strong consistency of $\widehat{\theta}_n$, along with almost sure convergence of the associated minimal distance. Moving to a limit distribution analysis, we characterize the high-dimensional limits of $\sqrt{n}(\widehat{\theta}_n - \theta^\star)$ and $\sqrt{n} \inf_{\theta \in \Theta} \mathsf{W}_1^{(\sigma)}(P_n, Q_\theta)$, thus establishing $n^{-1/2}$ convergence rates for both quantities in arbitrary dimension. Leveraging these results along with the framework from [14], we derive a high-dimensional generalization bound of order $n^{-1/2}$ on generative modeling with $\mathsf{W}_1^{(\sigma)}$. Empirical results to support our theory are provided. Using synthetic data we validate both the limiting distributions of parameter estimates and the convergence of the SWD as the number of samples increases.

Our main technical tool is a novel high-dimensional limit distribution result for scaled empirical SWD, i.e., $\sqrt{n}\mathsf{W}_1^{(\sigma)}(P_n, P)$, which may be of independent interest. Our analysis relies on the Kantorovich-Rubinstein (KR) duality for $\mathsf{W}_1$ [16], which allows representing $\sqrt{n}\mathsf{W}_1^{(\sigma)}(P_n, P)$ as a supremum of an empirical process indexed by the class of 1-Lipschitz functions convolved with a Gaussian density. We then prove that this function class is Donsker (i.e., satisfies the uniform central limit theorem (CLT)) under a polynomial moment condition on $P$.[5] By the continuous mapping theorem, we conclude that $\sqrt{n}\mathsf{W}_1^{(\sigma)}(P_n, P)$ converges in distribution to the supremum of a tight Gaussian process. To enable evaluation of the distributional limit, we also prove that the nonparametric bootstrap is consistent. The characterization of a high-dimensional limit distribution for empirical SWD stands in sharp contrast to the classic $\mathsf{W}_1$ case, for which such a result is known only when $d = 1$ [35].

## 1.2 Comparisons and Related Works

MDE questions similar to those studied herein were addressed for classic $\mathsf{W}_1$ in [36,37] (see also [38, 39]). They derived limit distribution results only for the one-dimensional case, essentially because it is unknown whether a properly scaled $\mathsf{W}_1(P_n, P)$ has a nondegenerate limit in general when $d > 1$.

Sliced Wasserstein distance MDE was recently analyzed in [40], covering arbitrary dimension, as done herein. Indeed, both sliced and smooth Wasserstein distances employ different approaches for alleviating the CoD. The sliced version eliminates dependence on $d$ by definition, as it is an average of one-dimensional $\mathsf{W}_1$ distances (via random projections of $d$-dimensional distributions). SWD, on

the other hand, does not entail dimensionality reduction, but leverages Gaussian smoothing to level out local irregularities in the high-dimensional distributions, which speeds up empirical convergence rates. This, in turn, enables a thorough MSWE asymptotic analysis for any $d$. Sliced and smooth Wasserstein distances are also similar in that they are both metrics and (topologically) equivalent to $\mathsf{W}_1$, but there are some notable differences. While sliced $\mathsf{W}_1$ is easily computable using the one-dimensional formula, computational aspects of SWD are still under exploration (see Section 6 for further discussion). SWD might be preferable as a proxy for regular $\mathsf{W}_1$, as the two are within an additive $2\sigma\sqrt{d}$ gap from one another [28, Lemma 1]. Comparison results for sliced Wasserstein seem weaker, assuming compact support and involving implicit dimension-dependent constants (cf., e.g., [41, Lemma 5.1.4]).

Also related to our work is entropic optimal transport (EOT). Its popularity has been driven both by algorithmic advances [42,43] (the latter gives a near-linear-time algorithm) and some statistical properties it possesses [44–46]. Specifically, two-sample empirical estimation under EOT is known to converge as $n^{-1/2}$ for smooth costs (thus, in particular, excluding entropic $\mathsf{W}_1$) with compactly supported distributions [47], or squared cost with subgassian distributions [48]. In comparison, SWD enjoys this fast convergence rate in the stronger one-sample setting and under milder conditions on the distribution. A CLT for empirical EOT under quadratic cost was also derived in [48]. This result is similar to that of [49] for the classic 2-Wasserstein distance, but is markedly different from ours. Notably, [48] derive the CLT with unknown centering constants given by the expected empirical EOT (which differs from the population one). Furthermore, unlike the SWD, EOT is not a metric, even when the underlying cost is [50, 51].[6] In conclusion, while EOT can be efficiently computed, several gaps are still present as far as its statistical properties, and perhaps more importantly, it surrenders some desirable structural properties of classic Wasserstein distances.

**Notation.** Let $\|\cdot\|$ denote the Euclidean norm, and $x \cdot y$, for $x, y \in \mathbb{R}^d$, designate the inner product. For any probability measure $Q$ on a measurable space $(S, \mathcal{S})$ and any measurable real function $f$ on $S$, we use the notation $Qf := \int_S f \, \mathrm{d}Q$ whenever the integral exists. We write $a \lesssim_x b$ when $a \leq C_x b$ for a constant $C_x$ that depends only on $x$ ($a \lesssim b$ means $a \leq Cb$ for an absolute constant $C$).

We denote by $(\Omega, \mathcal{A}, \mathbb{P})$ the underlying probability space on which all random variables are defined. The class of Borel probability measures on $\mathbb{R}^d$ is $\mathcal{P}(\mathbb{R}^d)$. The subset of measures with finite first moment is denoted by $\mathcal{P}_1(\mathbb{R}^d)$, i.e., $P \in \mathcal{P}_1(\mathbb{R}^d)$ whenever $\int \|x\| \, \mathrm{d}P(x) < \infty$. The convolution of $P, Q \in \mathcal{P}(\mathbb{R}^d)$ is $(P * Q)(\mathcal{A}) := \int \int \mathbb{1}_{\mathcal{A}}(x + y) \, \mathrm{d}P(x) \, \mathrm{d}Q(y)$, where $\mathbb{1}_{\mathcal{A}}$ is the indicator of $\mathcal{A}$. The convolution of measurable functions $f, g$ on $\mathbb{R}^d$ is $f * g(x) = \int f(x - y)g(y) \, \mathrm{d}y$. We also recall that $\mathcal{N}_\sigma := \mathcal{N}(0, \sigma^2 \mathrm{I}_d)$, and use $\varphi_\sigma(x) = (2\pi\sigma^2)^{-d/2} e^{-\|x\|^2/(2\sigma^2)}$, $x \in \mathbb{R}^d$, for the Gaussian density.

For a non-empty set $\mathcal{T}$, let $\ell^\infty(\mathcal{T})$ denote the space of all bounded functions $f : \mathcal{T} \to \mathbb{R}$, equipped with the uniform norm $\|f\|_{\mathcal{T}} := \sup_{t \in \mathcal{T}} |f(t)|$. We denote $\mathsf{Lip}_1(\mathbb{R}^d) := \{f : \mathbb{R}^d \to \mathbb{R} : |f(x) - f(y)| \leq \|x - y\| \; \forall x, y \in \mathbb{R}^d\}$ for the set of Lipschitz continuous functions on $\mathbb{R}^d$ with Lipschitz constant bounded by 1. When $d$ is clear from the context we use the shorthand $\mathsf{Lip}_1$.

## 2  Background and preliminaries

We next provide a short background on the central technical ideas used in the paper.

**1-Wasserstein distance.** The 1-Wasserstein distance $\mathsf{W}_1(P, Q)$ between $P, Q \in \mathcal{P}_1(\mathbb{R}^d)$ is

$$\mathsf{W}_1(P, Q) := \inf_{\pi \in \Pi(P, Q)} \int_{\mathbb{R}^d \times \mathbb{R}^d} \|x - y\| \, \mathrm{d}\pi(x, y),$$

where $\Pi(P, Q)$ is the set of all couplings of $P$ and $Q$. The KR duality further implies $\mathsf{W}_1(P, Q) = \sup_{f \in \mathsf{Lip}_1} \int_{\mathbb{R}^d} f \, \mathrm{d}(P - Q)$. See [16] for additional background.

**Empirical approximation.** Fix $P \in \mathcal{P}_1(\mathbb{R}^d)$ and let $X_1, \ldots, X_n \sim P$ be i.i.d. Let $P_n = n^{-1} \sum_{i=1}^n \delta_{X_i}$ be the empirical distribution of $X_1, \ldots, X_n$, where $\delta_x$ is the Dirac measure at $x$. The convergence rate of $\mathsf{W}_1(P_n, P)$ received much attention in the literature; see, e.g., [20, 52–58].[7]

Sharp rates are known in all dimensions;[8] $\mathbb{E}[\mathsf{W}_1(P_n, P)] = O(n^{-1/2})$ if $d = 1$, $= O(n^{-1/2} \log n)$ if $d = 2$, and $= O(n^{-1/d})$ for $d \geq 3$ provided that $P$ has sufficiently many moments (cf. [20]).

**Limit distribution.** Despite the comprehensive account of the expected $\mathsf{W}_1(P_n, P)$, limiting distribution results for a scaled version thereof are known only for $d = 1$. Indeed, Theorem 2 in [59] yields that $\mathsf{Lip}_1(\mathbb{R})$ is a $P$-Donsker class if (and only if) $\sum_j P([-j,j]^c)^{1/2} < \infty$. Combining with KR duality, we have $\sqrt{n}\mathsf{W}_1(P_n, P) \xrightarrow{d} \sup_{f \in \mathsf{Lip}_1(\mathbb{R})} G_P(f)$ for some tight Gaussian process $G_P$ in $\ell^\infty(\mathsf{Lip}_1(\mathbb{R}))$. An alternative derivation of the limit distribution for $d = 1$ is given in [35], based on the fact that $\mathsf{W}_1$ equals the $L^1$ distance between distribution functions when $d = 1$. The arguments in those papers, however, do not carry over to general $d$. For $d \geq 2$, in general, the function class $\mathsf{Lip}_1(\mathbb{R}^d)$ is not Donsker; if it was, then $\mathbb{E}[\mathsf{W}_1(P_n, P)]$ would be of order $O(n^{-1/2})$, contradicting existing results lower bounding the rate of convergence of $\mathsf{W}_1(P_n, P)$ [20].

**Smooth Wasserstein distance.** We are interested in $d \geq 2$, and instead of $\mathsf{W}_1$ consider the SWD [27, 28] $\mathsf{W}_1^{(\sigma)}(P, Q) := \mathsf{W}_1(P * \mathcal{N}_\sigma, Q * \mathcal{N}_\sigma)$. [27] shows that $\mathsf{W}_1^{(\sigma)}(P_n, P) = O_P(n^{-1/2})$, for all $d$ and any sub-Gaussian $P$. Herein, we characterize the limit distribution of $\sqrt{n}\mathsf{W}_1^{(\sigma)}(P_n, P)$, prove that this distribution can be accurately estimated via the bootstrap, and derive concentration inequalities (see Supplement A.1 for the latter). To simplify discussions, henceforth we assume $0 < \sigma \leq 1$.

**Stochastic processes.** A stochastic process $G := (G(t))_{t \in \mathcal{T}}$ indexed by $\mathcal{T}$ is Gaussian if the $(G(t_i))_{i=1}^k$ are jointly Gaussian for any finite collection $\{t_i\}_{i=1}^k \subset \mathcal{T}$. A Gaussian process $G$ is tight in $\ell^\infty(\mathcal{T})$ if and only if $\mathcal{T}$ is totally bounded for the pseudometric $d_G(s,t) = \sqrt{\mathbb{E}[|G(s) - G(t)|^2]}$, and $G$ has sample paths a.s. uniformly $d_G$-continuous [33, Section 1.5]. If $G$ is sample bounded, we view it as a mapping from the sample space into $\ell^\infty(\mathcal{T})$. A version of a stochastic process is another stochastic process with the same finite dimensional distributions.

## 3 Limit distribution theory for smooth Wasserstein distance

The main technical tool for treating MSWE is a characterization of the limit distribution of $\sqrt{n}\mathsf{W}_1^{(\sigma)}(P_n, P)$ in all dimensions, which is the focus of this section. We also derive consistency of the bootstrap as a means for computing the limit distribution, and establish concentration inequalities for $\mathsf{W}_1^{(\sigma)}(P_n, P)$ (see Supplement A.1 for the latter).

Starting from the limit distribution of $\sqrt{n}\mathsf{W}_1^{(\sigma)}(P_n, P)$, some definitions are needed to describe the limit random variable. Denote $\mathsf{Lip}_{1,0} := \{f \in \mathsf{Lip}_1 : f(0) = 0\}$, assume that $P\|x\|^2 < \infty$, and let $G_P^{(\sigma)} = (G_P^{(\sigma)}(f))_{f \in \mathsf{Lip}_{1,0}}$ be a centered Gaussian process with covariance function $\mathbb{E}[G_P^{(\sigma)}(f)G_P^{(\sigma)}(g)] = \mathsf{Cov}_P(f * \varphi_\sigma, g * \varphi_\sigma)$, where $f, g, \in \mathsf{Lip}_{1,0}$. One may verify that $|f * \varphi_\sigma(x)| \leq \|x\| + \sigma\sqrt{d}$ (cf. Section A.2), so that $P|f * \varphi_\sigma|^2 < \infty$, for all $f \in \mathsf{Lip}_{1,0}$ (which ensures that the covariance function above is well-defined). With that, we are ready to state the theorem.

**Theorem 1** (SWD limit distribution). *Assume that $P\|x\|^2 < \infty$. Let $\mathbb{R}^d = \bigcup_{j=1}^\infty I_j$ be a partition of $\mathbb{R}^d$ into bounded convex sets with nonempty interior such that $K := \sup_j \mathsf{diam}(I_j) < \infty$. If*

$$\sum_{j=1}^\infty M_j P(I_j)^{1/2} < \infty \quad \text{with} \quad M_j := \sup_{I_j} \|x\|, \tag{2}$$

*then there exists a version of $G_P^{(\sigma)}$ that is tight in $\ell^\infty(\mathsf{Lip}_{1,0})$, and denoting the tight version by the same symbol $G_P^{(\sigma)}$, we have $\sqrt{n}\mathsf{W}_1^{(\sigma)}(P_n, P) \xrightarrow{d} \sup_{f \in \mathsf{Lip}_{1,0}} G_P^{(\sigma)}(f) =: L_P^{(\sigma)}$. In addition, we have $\sqrt{n}\mathbb{E}\left[\mathsf{W}_1^{(\sigma)}(P_n, P)\right] \lesssim_{d,K} \sigma^{-\lfloor d/2 \rfloor} \sum_{j=1}^\infty M_j P(I_j)^{1/2}$.*

The proof is given in Supplement A.2. We use KR duality to translate the Gaussian convolution in the measure space to the convolution of Lipschitz functions with a Gaussian density. It is then shown that

this class of Gaussian-smoothed Lipschitz functions is $P$-Donsker by bounding the metric entropy of the function class restricted to each $I_j$. The proof substantially relies on empirical process theory.

**Remark 1** (Discussion on Condition (2)). *Let $\{I_j\}$ consist of cubes with side length 1 and integral lattice points as vertices. One may then obtain the bound*

$$\sum_{j=1}^{\infty} M_j P(I_j)^{1/2} \lesssim_d \sum_{k=1}^{\infty} k^d P(\|x\|_{\infty} > k)^{1/2} \lesssim \int_1^{\infty} t^d P(\|x\|_{\infty} > t)^{1/2}\, \mathrm{d}t,$$

*which is finite (by Markov's inequality) if there exists $\epsilon > 0$ such that $P|x_j|^{2(d+1)+\epsilon} < \infty$ for all $j$.*

*Proposition 1 in [27] shows that $\mathbb{E}\big[\mathsf{W}_1^{(\sigma)}(P_n, P)\big] = O(n^{-1/2})$ whenever $P$ is sub-Gaussian. Theorem 1 substantially relaxes this moment condition, in addition to deriving a limit distribution.*

**Remark 2** (Limit distribution for empirical $\mathsf{W}_p$). *The limit distribution of $\sqrt{n}\mathsf{W}_p(P_n, P)$, when $P$ is supported on a finite or a countable set, was derived in [60] and [61], respectively. [62] show asymptotic normality of $\sqrt{n}(\mathsf{W}_2(P_n, Q) - \mathbb{E}[\mathsf{W}_2(P_n, Q)])$, in arbitrary dimension, but under the assumption that $Q \neq P$. The limit distribution for the empirical 2-Wasserstein distance with $Q = P$ is known only when $d = 1$ [63]. None of the techniques employed in these works are applicable in our case, which therefore requires a different analysis as described above.*

The proof of Theorem 1 along with Lemma 3 from Supplement A.3 implies that the distribution of $L_P^{(\sigma)}$ can be estimated via the bootstrap [33, Chapter 3.6]. Let $X_1^B, \dots, X_n^B$ be i.i.d. from $P_n$ conditioned on $X_1, \dots, X_n$, and set $P_n^B := n^{-1}\sum_{i=1}^n \delta_{X_i^B}$ as the bootstrap empirical distribution. Let $\mathbb{P}^B$ be the probability measure induced by the bootstrap (i.e., the conditional probability given $X_1, X_2, \dots$).

**Corollary 1** (Bootstrap consistency). *Assume the conditions of Theorem 1 and that $P$ is not a point mass. Then, we have $\sup_{t \geq 0} \big|\mathbb{P}^B\big(\sqrt{n}\mathsf{W}_1^{(\sigma)}(P_n^B, P_n) \leq t\big) - \mathbb{P}\big(L_P^{(\sigma)} \leq t\big)\big| \to 0$ a.s.*

This corollary, together with continuity of the distribution function of $L_P^{(\sigma)}$ (cf. Lemma 3 in the Appendix), implies that for $\widehat{q}_{1-\alpha} := \inf\{t \geq 0 : \mathbb{P}^B\big(\sqrt{n}\mathsf{W}_1^{(\sigma)}(P_n^B, P_n) \leq t\big) \geq 1 - \alpha\}$ (which can be computed numerically), we have $\mathbb{P}\big(\sqrt{n}\mathsf{W}_1^{(\sigma)}(P_n, P) > \widehat{q}_{1-\alpha}\big) = \alpha + o(1)$.

**Remark 3** (Two-sample setting). *Theorem 1 and Corollary 1 can be extended to the two-sample case, i.e., accounting for $\mathsf{W}_1^{(\sigma)}(P_n, Q_m)$. The proof of Theorem 1 shows that the function class $\mathcal{F}_{\sigma,d} := \big\{f * \varphi_\sigma : f \in \mathsf{Lip}_{1,0}\big\}$ is Donsker, for all $d$. Consequently, $\sqrt{\frac{mn}{m+n}}\mathsf{W}_1^{(\sigma)}(P_n, Q_m)$ converges in distribution to the supremum of a tight Gaussian process, if the population distributions agree (cf. [64, Chapter 3.7]). By [64, Theorem 3.7.6], this limit process can be consistently estimated by the two-sample bootstrap. Two-sample testing with (unsmoothed) $\mathsf{W}_p$ is studied in [65], but they find critical values for the tests only for $d = 1$. This is due to lack of tractable distribution approximation results for $\mathsf{W}_p$ in high dimensions. Our theory shows that we can overcome this bottleneck by adopting $\mathsf{W}_1^{(\sigma)}$.*

## 4 Minimum Smooth Wasserstein Estimation

We study the statistical properties of the MSWE $\widehat{\theta}_n \in \operatorname{argmin}_{\theta \in \Theta} \mathsf{W}_1^{(\sigma)}(P_n, Q_\theta)$ in high dimensions. Here $P \in \mathcal{P}_1(\mathbb{R}^d)$, $P_n$ is the associated empirical measure, and $Q_\theta \in \mathcal{P}_1(\mathbb{R}^d)$, where $\theta \in \Theta \subset \mathbb{R}^{d_0}$, is the model class. We henceforth assume (without further mentioning) that the parameter space $\Theta \subset \mathbb{R}^{d_0}$ is compact with nonempty interior. The boundedness assumption on $\Theta$ can be weakened with some adjustments to the proofs of Theorems 2 and 3 below; cf. [37, Assumption 2.3].

### 4.1 Measurability and Consistency

The following theorem states that the MSWE is measurable. The proof (given in Supplement B.2) relies on Corollary 1 in [66], which provides a sufficient condition for the desired measurability.

**Theorem 2** (MSWE measurability). *Assume that the map $\theta \mapsto Q_\theta$ is continuous relative to the weak topology,[9] i.e., $Q_\theta \rightharpoonup Q_{\bar{\theta}}$ whenever $\theta \to \bar{\theta}$ in $\Theta$. Then, for every $n \in \mathbb{N}$, there exists a measurable function $\omega \mapsto \widehat{\theta}_n(\omega)$ such that $\widehat{\theta}_n(\omega) \in \operatorname{argmin}_{\theta \in \Theta} \mathsf{W}_1^{(\sigma)}\big(P_n(\omega), Q_\theta\big)$ for every $\omega \in \Omega$ (this also implies that $\operatorname{argmin}_{\theta \in \Theta} \mathsf{W}_1^{(\sigma)}\big(P_n(\omega), Q_\theta\big)$ is nonempty).*

Next, we establish consistency of the MSWE. The proof relies on [67, Theorem 7.33]. To apply it, we verify epi-convergence of the map $\theta \mapsto \mathsf{W}_1^{(\sigma)}(P_n, Q_\theta)$ towards $\theta \mapsto \mathsf{W}_1^{(\sigma)}(P, Q_\theta)$ (after proper extensions). See Supplement B.3 for details.

**Theorem 3** (MSWE consistency). *Assume that the map $\theta \mapsto Q_\theta$ is continuous relative to the weak topology. Then, we have $\inf_{\theta \in \Theta} \mathsf{W}_1^{(\sigma)}(P_n, Q_\theta) \to \inf_{\theta \in \Theta} \mathsf{W}_1^{(\sigma)}(P, Q_\theta)$ a.s. In addition, there exists an event with probability one on which the following holds: for any sequence $\{\widehat{\theta}_n\}_{n \in \mathbb{N}}$ of measurable estimators such that $\mathsf{W}_1^{(\sigma)}(P_n, Q_{\widehat{\theta}_n}) \leq \inf_{\theta \in \Theta} \mathsf{W}_1^{(\sigma)}(P_n, Q_\theta) + o(1)$, the set of cluster points of $\{\widehat{\theta}_n\}_{n \in \mathbb{N}}$ is included in $\operatorname{argmin}_{\theta \in \Theta} \mathsf{W}_1^{(\sigma)}(P, Q_\theta)$. In particular, if $\operatorname{argmin}_{\theta \in \Theta} \mathsf{W}_1^{(\sigma)}(P, Q_\theta)$ is unique, i.e., $\operatorname{argmin}_{\theta \in \Theta} \mathsf{W}_1^{(\sigma)}(P, Q_\theta) = \{\theta^\star\}$, then $\widehat{\theta}_n \to \theta^\star$ a.s.*

## 4.2 Limit Distributions

We study the limit distributions of the MSWE and the associated SWD. Results are presented for the 'well-specified' setting, i.e., when $P = Q_{\theta^\star}$ for some $\theta^\star$ in the interior of $\Theta \subset \mathbb{R}^{d_0}$. Extensions to the 'misspecified' case are straightforward (cf. [37, Theorem B.8]). Our derivation leverages the method of [2] for MDE analysis over normed spaces. To make the connection, we need some definitions.

For any $G \in \ell^\infty(\mathsf{Lip}_{1,0})$, define $\|G\|_{\mathsf{Lip}_{1,0}} := \sup_{f \in \mathsf{Lip}_{1,0}} |G(f)|$. With any $Q \in \mathcal{P}_1(\mathbb{R}^d)$, associate the functional $Q^{(\sigma)} : \mathsf{Lip}_{1,0} \to \mathbb{R}$ defined by $Q^{(\sigma)}(f) := Q(f * \varphi_\sigma) = (Q * \mathcal{N}_\sigma)(f)$. Note that $\|Q^{(\sigma)}\|_{\mathsf{Lip}_{1,0}} := \sup_{f \in \mathsf{Lip}_{1,0}} |Q^{(\sigma)}(f)|$ is finite as $Q \in \mathcal{P}_1(\mathbb{R}^d)$ and $|(f * \varphi_\sigma)(x)| \leq \|x\| + \sigma\sqrt{d}$ for any $f \in \mathsf{Lip}_{1,0}$. Consequently, $Q^{(\sigma)} \in \ell^\infty(\mathsf{Lip}_{1,0})$ for any $Q \in \mathcal{P}_1(\mathbb{R}^d)$. Finally, observe that $\mathsf{W}_1^{(\sigma)}(P, Q) = \|P^{(\sigma)} - Q^{(\sigma)}\|_{\mathsf{Lip}_{1,0}}$, for any $P, Q \in \mathcal{P}_1(\mathbb{R}^d)$ (cf. Supplement A.2).

**SWD limit distribution.** We start from the limit distribution of the (scaled) infimized SWD. This result is central for deriving the limiting MSWE distribution (see Theorem 5 and Corollary 2 below). Theorem 4 is proven in Supplement B.4 via an adaptation of the argument from [2, Theorem 4.2].

**Theorem 4** (Minimal SWD limit distribution). *Let $P$ satisfy the conditions of Theorem 1. In addition, suppose that (i) the map $\theta \mapsto Q_\theta$ is continuous relative to the weak topology; (ii) $P \neq Q_\theta$ for any $\theta \neq \theta^\star$; (iii) there exists a vector-valued functional $D^{(\sigma)} \in (\ell^\infty(\mathsf{Lip}_{1,0}))^{d_0}$ such that $\|Q_\theta^{(\sigma)} - Q_{\theta^\star}^{(\sigma)} - \langle \theta - \theta^\star, D^{(\sigma)} \rangle\|_{\mathsf{Lip}_{1,0}} = o(\|\theta - \theta^\star\|)$ as $\theta \to \theta^\star$, where $\langle t, D^{(\sigma)} \rangle := \sum_{i=1}^{d_0} t_i D_i^{(\sigma)}$ for $t \in \mathbb{R}^{d_0}$; (iv) the derivative $D^{(\sigma)}$ is nonsingular in the sense that $\langle t, D^{(\sigma)} \rangle \neq 0$, i.e., $\langle t, D^{(\sigma)} \rangle \in \ell^\infty(\mathsf{Lip}_{1,0})$ is not the zero functional for all $0 \neq t \in \mathbb{R}^{d_0}$. Then, $\sqrt{n} \inf_{\theta \in \Theta} \mathsf{W}_1^{(\sigma)}(P_n, Q_\theta) \xrightarrow{d} \inf_{t \in \mathbb{R}^{d_0}} \|G_P^{(\sigma)} - \langle t, D^{(\sigma)} \rangle\|_{\mathsf{Lip}_{1,0}}$, where $G_P^{(\sigma)}$ is the Gaussian process from Theorem 1.*

**Remark 4** (Norm differentiability). *Condition (iii) in Theorem 4 is called 'norm differentiability' in [2]. In these terms, the theorem assumes that the map $\theta \mapsto Q_\theta^{(\sigma)}, \Theta \to \ell^\infty(\mathsf{Lip}_{1,0})$, is norm differentiable around $\theta^\star$ with derivative $D^{(\sigma)}$. This allows approximating the map $\theta \mapsto Q_\theta^{(\sigma)}$ by the affine function $Q_{\theta^\star}^{(\sigma)} + \langle \theta - \theta^\star, D^{(\sigma)} \rangle$ near $\theta^\star$. Together with the result of Theorem 1 and the right reparameterization, norm differentiability is key for establishing the theorem.*

**Remark 5** (Primitive conditions for norm differentiability). *Suppose that $\{Q_\theta\}_{\theta \in \Theta}$ is dominated by a common Borel measure $\nu$ on $\mathbb{R}^d$, and let $q_\theta$ denote the density of $Q_\theta$ with respect to $\nu$, i.e., $\mathsf{d}Q_\theta = q_\theta \, \mathsf{d}\nu$. Then, $Q_\theta * \mathcal{N}_\sigma$ has Lebesgue density $x \mapsto \int \varphi_\sigma(x-t) q_\theta(x) \, \mathsf{d}\nu(t)$. Assume that $q_\theta$ admits the Taylor expansion $q_\theta(x) = q_{\theta^\star}(x) + \dot{q}_{\theta^\star}(x) \cdot (\theta - \theta^\star) + r_\theta(x) \cdot (\theta - \theta^\star)$ with $r_\theta(x) = o(1)$ as $\theta \to \theta^\star$.*

*Then, one may verify that Condition (iii) holds with $D^{(\sigma)}(f) = \int f(x) \int \varphi_\sigma(x-t)\dot{q}_{\theta^\star}(t)\,\mathrm{d}\nu(t)\,\mathrm{d}x = \int (f * \varphi_\sigma)(t)\dot{q}_{\theta^\star}(t)\,\mathrm{d}\nu(t)$, for $f \in \mathsf{Lip}_{1,0}$, provided that $\int(1+\|t\|)\|\dot{q}_{\theta^\star}(t)\|\,\mathrm{d}\nu(t) < \infty$ and $\int(1+\|t\|)\|r_\theta(t)\|\,\mathrm{d}\nu(t) = o(1)$ (use the fact that $|f(t)| \le \|t\|$, for any $f \in \mathsf{Lip}_{1,0}$).*

**MSWE limit distribution.** We study convergence in distribution of the MSWE. Optimally, the limit distribution of $\sqrt{n}(\widehat{\theta}_n - \theta^\star)$, for some $\widehat{\theta}_n \in \mathrm{argmin}_{\theta\in\Theta} \mathsf{W}_1^{(\sigma)}(P_n, Q_\theta)$, is the object of interest. However, a limit is guaranteed to exist only when the (convex) function $t \mapsto \big\|G_P^{(\sigma)} - \langle t, D^{(\sigma)}\rangle\big\|_{\mathsf{Lip}_{1,0}}$ has a unique minimum a.s. (see Corollary 2 below). To avoid this stringent assumption, before treating $\sqrt{n}(\widehat{\theta}_n - \theta^\star)$, we first consider the set of approximate minimizers $\widehat{\Theta}_n := \big\{\theta \in \Theta : \mathsf{W}_1^{(\sigma)}(P_n, Q_\theta) \le \inf_{\theta'\in\Theta} \mathsf{W}_1^{(\sigma)}(P_n, Q_{\theta'}) + \lambda_n/\sqrt{n}\big\}$, where $\{\lambda_n\}_{n\in\mathbb{N}}$ is an arbitrary $o_\mathbb{P}(1)$ sequence.

We show that $\widehat{\Theta}_n \subset \theta^\star + n^{-1/2}K_n$ for some (random) sequence of compact convex sets $\{K_n\}_{n\in\mathbb{N}}$ with inner probability approaching one. Resorting to inner probability seems inevitable since the event $\{\widehat{\Theta}_n \subset \theta^\star + n^{-1/2}K_n\}$ need not be measurable in general (see [2, Section 7]). To define such sequence $\{K_n\}_{n\in\mathbb{N}}$, for any $L \in \ell^\infty(\mathsf{Lip}_{1,0})$ and $\beta \ge 0$, let $K(L,\beta) := \Big\{t \in \mathbb{R}^{d_0} : \big\|L - \langle t, D^{(\sigma)}\rangle\big\|_{\mathsf{Lip}_{1,0}} \le \inf_{t'\in\mathbb{R}^d} \big\|L - \langle t', D^{(\sigma)}\rangle\big\|_{\mathsf{Lip}_{1,0}} + \beta\Big\}$. Lemma 7.1 of [2] ensures that for any $\beta \ge 0$, $L \mapsto K(L,\beta)$ is a measurable map from $\ell^\infty(\mathsf{Lip}_{1,0})$ into $\mathfrak{K}$ – the class of all compact, convex, and nonempty subsets of $\mathbb{R}^{d_0}$ – endowed with the Hausdorff topology. That is, the topology induced by the metric $d_\mathsf{H}(K_1, K_2) := \inf\big\{\delta > 0 : K_1^\delta \supset K_2, K_2^\delta \supset K_1\big\}$, where $K^\delta := \bigcup_{x\in K}\big\{y \in \mathbb{R}^{d_0} : \|y - x\| \le \delta\big\}$ is the $\delta$-blowup of $K$.

**Theorem 5** (MSWE limit distribution). *Under the conditions of Theorem 4, there exists a sequence of nonnegative reals $\beta_n \downarrow 0$ such that (i) $\mathbb{P}_*\big(\widehat{\Theta}_n \subset \theta^\star + n^{-1/2}K(\mathbb{G}_n^{(\sigma)}, \beta_n)\big) \to 1$, where $\mathbb{G}_n^{(\sigma)} := \sqrt{n}(P_n^{(\sigma)} - P^{(\sigma)})$ is the (smooth) empirical process and $\mathbb{P}_*$ denotes inner probability; and (ii) $K(\mathbb{G}_n^{(\sigma)}, \beta_n) \xrightarrow{d} K(G_P^{(\sigma)}, 0)$ as $\mathfrak{K}$-valued random variables.*

Given Theorem 4, the proof of Theorem 5 follows by a verbatim repetition of the argument from [2, Section 7.2]. The details are therefore omitted. If $\mathrm{argmin}_{t\in\mathbb{R}^{d_0}} \big\|G_P^{(\sigma)} - \langle t, D^{(\sigma)}\rangle\big\|_{\mathsf{Lip}_{1,0}}$ is unique a.s. (a nontrivial assumption), then Theorem 5 simplifies as follows.[10]

**Corollary 2** (Simplified MSWE limit distribution). *Assume the conditions of Theorem 4. Let $\{\widehat{\theta}_n\}_{n\in\mathbb{N}}$ be a sequence measurable estimators such that $\mathsf{W}_1^{(\sigma)}(P_n, Q_{\widehat{\theta}_n}) \le \inf_{\theta\in\Theta} \mathsf{W}_1^{(\sigma)}(P_n, Q_\theta) + o_\mathbb{P}(n^{-1/2})$. Then, provided that $\mathrm{argmin}_{t\in\mathbb{R}^{d_0}} \big\|G_P^{(\sigma)} - \langle t, D^{(\sigma)}\rangle\big\|_{\mathsf{Lip}_{1,0}}$ is unique a.s., we have*
$$\sqrt{n}(\widehat{\theta}_n - \theta^\star) \xrightarrow{d} \mathrm{argmin}_{t\in\mathbb{R}^{d_0}} \big\|G_P^{(\sigma)} - \langle t, D^{(\sigma)}\rangle\big\|_{\mathsf{Lip}_{1,0}}.$$

Corollary 2 is proven in Supplement B.5 using weak convergence of argmin of random convex maps.

**Generalization discussion.** Theorem 5 and Corollary 2 imply a generalization bound for MSWE-based generative model. Focusing on the corollary, $\widehat{\theta}_n$ is an approximate optimizer (e.g., obtained via some suboptimal gradient-based optimization) of the empirical MSWE problem $\mathsf{W}_1^{(\sigma)}(P_n, Q_\theta)$. The goal is to obtain a $\widehat{\theta}_n$ that approximates as best as possible the minimizer $\theta^\star$ of the population loss $\mathsf{W}_1^{(\sigma)}(P, Q_\theta)$. Corollary 2 thus states that the MSWE $\widehat{\theta}_n$ converges to the true optimum $\theta^\star$ (in expectation, in distribution, and with high probability) at a dimension-free rate of $n^{-1/2}$, which corresponds to the notion of GAN generalization from [9, 14]. In fact, by Eq. (10) from [14], we further have that $\mathsf{W}_1^{(\sigma)}(P, Q_{\widehat{\theta}_n}) - \inf_{\theta\in\Theta} \mathsf{W}_1^{(\sigma)}(P, Q_\theta) \lesssim n^{-1/2}$, with high probability.

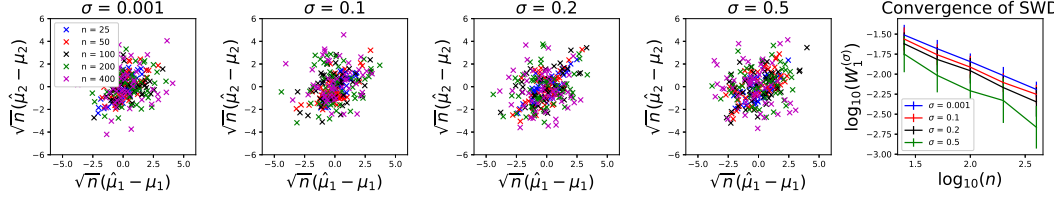

Figure 1: One-dimensional limiting distributions for the two mean parameters of the mixture $P = 0.5\mathcal{N}(\mu_1, 1) + 0.5\mathcal{N}(\mu_2, 1)$, for $\mu_1 = 0$ and $\mu_2 = 1$. Also shown on a log-log scale (with error bars) is the SWD convergence as a function of $n$.

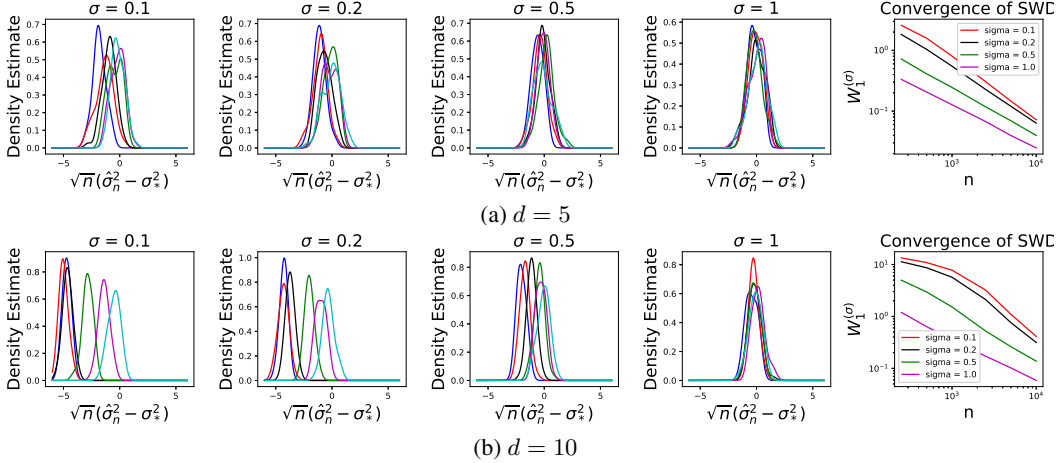

(a) $d = 5$

(b) $d = 10$

Figure 2: Empirical limiting distributions for the variance parameter of an MSWE-based generative model fitted to $P = \mathcal{N}_1$. Also shown as a log-log plot is the SWD convergence as a function of $n$.

## 5 Empirical Results

We provide experiments on synthetic data validating our theory. We start with a one-dimensional setting since then an exact expression for the SWD is available (as the $L^1$ distance between cumulative distribution functions [68]). Afterwards, higher dimensional problems are explored using an estimator based on the neural network (NN) parameterized KR dual form of $W_1$.

Fig. 1 shows results for fitting two-parameter generative models in one dimension for a Gaussian mixture (parameterized by the two means, one from each mode). $\sqrt{n}$-scaled scatter plots of the estimation error are shown for various $\sigma$ and $n$ values, each formed from 50 estimation trials. Convergence of the corresponding SWD losses is shown on the right. Note that the point clouds closely overlap in each plot even as $n$ increases, implying that indeed a limiting distribution is emerging, as predicted by the theory. In particular, the spread of the scatter plots does not increase despite a 16-fold increase in $n$, and the SWD loss converges at approximately an $n^{-1/2}$ rate. Supplement C gives additional results for a single Gaussian (parameterized by mean and variance).

In higher dimensions, the MSWE is computed as follows. We first draw $n$ samples from $P$ and obtain the empirical measure $P_n$. Sampling from $P_n$ and $\mathcal{N}_\sigma$ and adding the obtained values produces samples from $P_n * \mathcal{N}_\sigma$. Applying similar steps to $Q_\theta$, we may compute $W_1^{(\sigma)}(P_n, Q_\theta)$ by applying standard $W_1$ estimators to samples from the convolved measures. We use the NN-based estimator for WGAN-GP discriminator from [29]. As a side note, we believe that more effective estimators that are tailored for the SWD structure are possible, but leave this exploration to future work.

Fig. 2 shows MSWE results in dimensions 5 and 10. The target distribution is a multivariate standard Gaussian $P = \mathcal{N}_{\sigma_\star}$ for $\sigma_\star = 1$. The model $Q_\theta$ is also an isotropic Gaussian, with a single (variance) parameter. The WGAN-GP discriminator has 3 hidden layers with 512 hidden units each. The resulting distribution of $\sqrt{n}(\widehat{\sigma}_n^2 - \sigma_\star^2)$ is shown for various values of $\sigma$ (the SWD smoothing parameter) and number of samples $n$. These distributions are computed using a kernel density esti-

mate on 50 random trials. As seen in the figure, the distribution of $\sqrt{n}(\widehat{\sigma}_n^2 - \sigma_\star^2)$ converges to a clear limit as $n$ increases, for all $\sigma > 0$ values (although convergence is slower for smaller $\sigma$). When $\sigma$ is smaller, i.e., closer to the classic $W_1$ case, convergence is less pronounced, especially in higher dimensions. Finally, note that as predicted by our theory, the MSWE convergence rate is $n^{-1/2}$.

Lastly, we consider a more complex target $P$ and parameterize $Q_\theta$ via a three-layer neural network with 256 hidden units per layer. Note that this corresponds to the generator in a GAN setup, where the parameters $\theta$ of the neural network are learned so that $Q_\theta$ matches a target distribution. We combine this with our neural SWD-based discriminator, effectively creating an SWD GAN that we train in a way similar to WGAN-GP. Setting $d = 10$, we take $P$ as a $2^d$-mode Gaussian mixture formed by equal-weighted isotropic Gaussians (with variance parameter 1) centered at each corner of the $[-1, 1]^d$ hypercube. As there are too many parameters to visualize the limiting distribution, Fig. 3 instead shows the SWD convergence versus the number of samples $n$. As predicted, for $\sigma > 0$, the SWD asymptotically converges as approximately $n^{-1/2}$, though for smaller $\sigma$ this

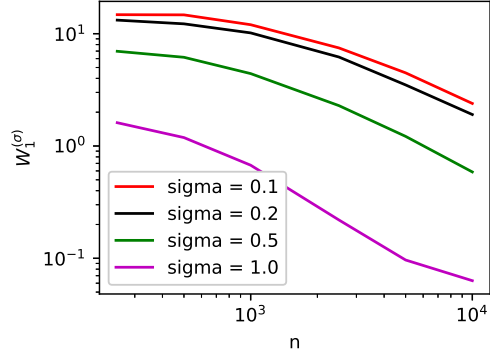

Figure 3: Convergence for fitting an NN generative model to a multivariate Gaussian mixture.

rate only kicks in for higher values of $n$. This two-phase behavior is expected, since when $n$ and $\sigma$ are small, and $d$ is large, the Gaussian convolution in the SWD is unlikely to result in smoothing different samples together.

## 6   Summary and concluding remarks

We studied MDE with $W_1^{(\sigma)}$ as the figure of merit. Measurability, strong consistency and limit distributions for MSWE in arbitrary dimensions were established. The characterization of high-dimensional distributional limits stands in sharp contrast with the classic $W_1$ MDE, where such a result is known only for $d = 1$ [37]. In particular, our results imply a uniform $n^{-1/2}$ convergence rate for MSWE for all $d$, highlighting the virtue of $W_1^{(\sigma)}$ for high-dimensional generative modeling. Our ability to treat MSWE for arbitrary $d$ relied on a novel limit distribution result for the empirical SWD. Under a polynomial moment condition on $P$ we show that $\sqrt{n}W_1^{(\sigma)}(P_n, P)$ converges in distribution to the supremum of a tight Gaussian process. This again contrasts the $W_1$ case, where a limit distribution is known only in $d = 1$. We have also established consistency of the bootstrap to enable evaluation of the limit distribution in practice.

This work focuses of statistical aspects of generative modeling with the SWE. A major goal going forward is to develop efficient algorithms for computing $W_1^{(\sigma)}$, that are tailored to exploit the Gaussian convolution structure. We view the Monte Carlo algorithm employed herein merely as a placeholder. Gaussian smoothing significantly speeds up $W_1$ empirical convergence rates from $n^{-1/d}$ to $\sigma^{-d/2}n^{-1/2}$. While the latter is optimal in $n$, the exponential dependence of the prefactor on $d$ calls for further exploration. We aim to relax this dependence under the manifold hypothesis, showing that the actual dependence is on the intrinsic dimension, and not the ambient one. Additional directions include an analysis for when $\sigma \downarrow 0$ is at a sufficiently slow rate (as a proxy for $W_1$). This is both theoretically challenging (calls for a finer analysis than the one presented herein) and practically relevant, as noise annealing is often used to stabilize training. SWDs of higher orders are also of interest.

## Broader Impact

Our goal is to provide a stronger theoretical foundation for generative modeling based on the smoothed Wasserstein distance. We hope that this enables practitioners to build more robust, fair, and resilient generative models.

## Funding Disclosure

The work of Z. Goldfeld was supported by the National Science Foundation Grant CCF-1947801 and the 2020 IBM Faculty Award. The work of K. Kato was supported by the National Science Foundation Grants DMS-1952306 and DMS-2014636.

## Footnotes

[1] Recall that $\delta$ is an SD if $\delta(P, Q) = 0 \iff P = Q$.

[2] or a variant thereof, where $Q_\theta$ is also estimated from samples.

[3] One might hope that using more sophisticated estimates of $P$ (instead of the empirical measure) or avoiding plugin methods altogether may alleviate the CoD. However, recent minimax analyses for Wasserstein distances [24], $f$-divergences [25] and integral probability metrics [26] show that the $n^{-1/d}$ rate is generally unavoidable.

[4] The explicit bound from [31] is $\mathbb{E}\big[\delta^{(\sigma)}(P_n, P)\big] \lesssim \sigma^{-d/2} n^{-1/2}$. While the dependence on $n$ is optimal and decoupled from $d$ (unlike in CoD rates), the prefactor is exponential in $d$—a dependence that warrants further exploration. See discussion in Section 6.

[5] The reader is referred to, e.g., [32–34] as useful references on modern empirical process theory.

[6]EOT can be transformed into a Sinkhorn divergence via a simple modification, but it is still is not a metric since it lacks the triangle inequality [51].

[7]Those references also contain results on the more general Wasserstein distance and non-Euclidean spaces.

[8]Except $d = 2$, where a log factor is possibly missing.

[9]The *weak topology* on $\mathcal{P}(\mathbb{R}^d)$ is induced by integration against the set $C_b(\mathbb{R}^d)$ of bounded and continuous functions, i.e., $(\mu_k)_{k \in \mathbb{N}}$ converges weakly to $\mu$, denoted by $\mu_k \rightharpoonup \mu$, if $\mu_k(f) \to \mu(f)$, for all $f \in C_b(\mathbb{R}^d)$.

[10]Note that $\mathrm{argmin}_{t\in\mathbb{R}^{d_0}} \big\|G_P^{(\sigma)} - \langle t, D^{(\sigma)}\rangle\big\|_{\mathsf{Lip}_{1,0}} \ne \emptyset$ provided that $D^{(\sigma)}$ is nonsingular, since the latter guarantees that $\big\|G_P^{(\sigma)} - \langle t, D^{(\sigma)}\rangle\big\|_{\mathsf{Lip}_{1,0}} \to \infty$ as $\|t\| \to \infty$.

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
