[Supplementary Material]

# Supplementary Material: Asymptotic Guarantees for Generative Modeling based on the Smooth Wasserstein Distance

## A  Additional result and proofs for Section 2

### A.1  Concentration inequalities for $\mathsf{W}_1^{(\sigma)}(P_n, P)$

We consider a quantitative concentration inequality for $\mathsf{W}_1^{(\sigma)}(P_n, P)$. For $\alpha > 0$, let $\|\xi\|_{\psi_\alpha} := \inf\{C > 0 : \mathbb{E}[e^{(|\xi|/C)^\alpha}] \leq 2\}$ be the Orlitz $\psi_\alpha$-norm for a real-valued random variable $\xi$ (if $\alpha \in (0,1)$, then $\|\cdot\|_{\psi_\alpha}$ is a quasi-norm). In Section A.4 we prove the following.

**Corollary 3** (Concentration inequality). *Assume* $\mathbb{E}[\mathsf{W}_1^{(\sigma)}(P_n, P)] < \infty$. *The following hold:*

*(i) If $P$ is compactly supported with support $\mathcal{X}$, then*

$$\mathbb{P}\left(\mathsf{W}_1^{(\sigma)}(P_n, P) \geq \mathbb{E}[\mathsf{W}_1^{(\sigma)}(P_n, P)] + t\right) \leq e^{-\frac{nt^2}{\mathrm{diam}(\mathcal{X})^2}}, \quad \forall t > 0.$$

*(ii) If $\big\|\|X\|\big\|_{\psi_\alpha} < \infty$ for some $\alpha \in (0,1]$, where $X \sim P$, then for any $\eta > 0$, there exists a constant $C = C_{\eta,\alpha}$ depending only on $\eta, \alpha$ such that*

$$\mathbb{P}\left(\mathsf{W}_1^{(\sigma)}(P_n, P) \geq (1+\eta)\mathbb{E}[\mathsf{W}_1^{(\sigma)}(P_n, P)] + t\right) \leq \exp\left(-\frac{nt^2}{C(P\|x\|^2 + \sigma^2 d)}\right)$$

$$+ 3\exp\left(-\left(\frac{nt}{C\left(\big\|\max_{1 \leq i \leq n}\|X_i\|\big\|_{\psi_\alpha} + \sigma\sqrt{d}\right)}\right)^\alpha\right), \quad \forall t > 0.$$

*(iii) If $P\|x\|^q < \infty$ for some $q \in [1, \infty)$, then for any $\eta > 0$, there exists a constant $C = C_{\eta,q}$ depending only on $\eta, q$ such that*

$$\mathbb{P}\left(\mathsf{W}_1^{(\sigma)}(P_n, P) \geq (1+\eta)\mathbb{E}[\mathsf{W}_1^{(\sigma)}(P_n, P)] + t\right) \leq \exp\left(-\frac{nt^2}{C(P\|x\|^2 + \sigma^2 d)}\right)$$

$$+ \frac{C\left(\mathbb{E}[\max_{1 \leq i \leq n}\|X_i\|^q] + \sigma^q d^{q/2}\right)}{n^q t^q}, \quad \forall t > 0.$$

### A.2  Proof of Theorem 1

Recall that $\varphi_\sigma$ is the density function of $\mathcal{N}(0, \sigma^2 \mathrm{I}_d)$, i.e., $\varphi_\sigma(x) = (2\pi\sigma^2)^{-d/2}e^{-\|x\|^2/(2\sigma^2)}$ for $x \in \mathbb{R}^d$. Noting that the measure $P_n * \mathcal{N}_\sigma$ has density

$$x \mapsto \frac{1}{n}\sum_{i=1}^n \varphi_\sigma(x - X_i) = \frac{1}{n}\sum_{i=1}^n \varphi_\sigma(X_i - x),$$

we arrive at the expression

$$\mathsf{W}_1^{(\sigma)}(P_n, P) = \sup_{f \in \mathsf{Lip}_1}\left[\frac{1}{n}\sum_{i=1}^n f * \varphi_\sigma(X_i) - Pf * \varphi_\sigma\right]. \tag{3}$$

The RHS of (3) does not change even if we replace $f$ by $f - f(x^\star)$ for any fixed point $x^\star$ (as $\int_{\mathbb{R}^d} \varphi_\sigma(x^\star - y)dy = 1$). Thus, the problem boils down to showing that the function class

$$\check{\mathcal{F}} := \check{\mathcal{F}}_{\sigma,d} := \{f * \varphi_\sigma : f \in \mathsf{Lip}_{1,0}\} \quad \text{with } \mathsf{Lip}_{1,0} := \{f \in \mathsf{Lip}_1 : f(0) = 0\}$$

is $P$-Donsker. Pick any $f \in \mathsf{Lip}_{1,0}$, and consider

$$f_\sigma(x) := f * \varphi_\sigma(x) = \int f(y)\varphi_\sigma(x - y)\,\mathsf{d}y.$$

We see that, since $|f(y)| \leq |f(0)| + \|y\| = \|y\|$,

$$
|f_\sigma(x)| \leq \int \|y\| \varphi_\sigma(x-y) \, \mathsf{d}y \leq \int (\|x\| + \|x-y\|) \varphi_\sigma(x-y) \, \mathsf{d}y
$$

$$
\leq \|x\| + \int \|y\| \varphi_\sigma(y) \, \mathsf{d}y \leq \|x\| + \left( \int_{\mathbb{R}^d} \|y\|^2 \varphi_\sigma(y) \, \mathsf{d}y \right)^{1/2}
$$

$$
= \|x\| + \sigma\sqrt{d}.
$$

In general, for a vector $k = (k_1, \ldots, k_d)$ of $d$ nonnegative integers, define the differential operator

$$
D^k = \frac{\partial^{|k|}}{\partial x_1^{k_1} \cdots \partial x_d^{k_d}},
$$

with $|k| = \sum_{i=1}^d k_i$. We next give a uniform bound on the derivatives of $f_\sigma$, for any $f \in \mathsf{Lip}_1$.

**Lemma 1** (Uniform bound on derivatives)**.** *For any $f \in \mathsf{Lip}_1$ and any nonzero multiindex $k = (k_1, \ldots, k_d)$, we have*

$$
\left| D^k f_\sigma(x) \right| \leq \sigma^{-|k|+1} \sqrt{(|k|-1)!}, \quad \forall x \in \mathbb{R}^d.
$$

*Proof.* Let $H_m(z)$ denote the Hermite polynomial of degree $m$ defined by

$$
H_m(z) = (-1)^m e^{z^2/2} \left[ \frac{d^m}{dz^m} e^{-z^2/2} \right], \quad m = 0, 1, \ldots.
$$

Note that for $Z \sim \mathcal{N}(0,1)$, $\mathbb{E}[H_m(Z)^2] = m!$.

A straightforward computation shows that

$$
D_x^k \varphi_\sigma(x-y) = \varphi_\sigma(x-y) \left[ \prod_{j=1}^d (-1)^{k_j} \sigma^{-k_j} H_{k_j} \big( (x_j - y_j)/\sigma \big) \right]
$$

for any multiindex $k = (k_1, \ldots, k_d)$, where $D_x$ means that the differential operator is applied to $x$. Hence, we have

$$
D^k f_\sigma(x) = \int f(y) \varphi_\sigma(x-y) \left[ \prod_{j=1}^d (-1)^{k_j} \sigma^{-k_j} H_{k_j} \big( (x_j - y_j)/\sigma \big) \right] \mathsf{d}y
$$

$$
= \int f(x - \sigma y) \varphi_1(y) \left[ \prod_{j=1}^d (-1)^{k_j} \sigma^{-k_j} H_{k_j}(y_j) \right] \mathsf{d}y,
$$

so that, by 1-Lipschitz continuity of $f$,

$$
\left| D^k f_\sigma(x) - D^k f_\sigma(x') \right| \leq \|x - x'\| \int \varphi_1(y) \left[ \prod_{j=1}^d \sigma^{-k_j} \left| H_{k_j}(y_j) \right| \right] \mathsf{d}y.
$$

Note that the integral on the RHS equals

$$
\prod_{j=1}^d \sigma^{-k_j} \mathbb{E}\big[ \left| H_{k_j}(Z) \right| \big] \leq \prod_{j=1}^d \sigma^{-k_j} \sqrt{\mathbb{E}\left[ \left| H_{k_j}(Z) \right|^2 \right]} = \prod_{j=1}^d \sigma^{-k_j} \sqrt{k_j!} \leq \sigma^{-|k|} \sqrt{|k|!},
$$

where $Z \sim \mathcal{N}(0,1)$. The conclusion of the lemma follows from induction on the size of $|k|$. $\quad\square$

We will use the following technical result.

**Lemma 2** (Metric entropy bound for Hölder ball). *Let $\mathcal{X}$ be a bounded convex subset of $\mathbb{R}^d$ with nonempty interior. For given $N \in \mathbb{N}$ and $M > 0$, let $C^N(\mathcal{X})$ be the set of continuous real functions on $\mathcal{X}$ that are $N$-times differentiable on the interior of $\mathcal{X}$, and consider the Hölder ball with smoothness $N$ and radius $M$*

$$C_M^N(\mathcal{X}) := \left\{ f \in C^N(\mathcal{X}) : \|f\|_{C^N(\mathcal{X})} \le M \right\},$$

*where $\|f\|_{C^N(\mathcal{X})} := \max_{0 \le |k| \le N} \sup_x |D^k f(x)|$ (the suprema are taken over the interior of $\mathcal{X}$). Then, the metric entropy of $C_M^N(\mathcal{X})$ (w.r.t. the uniform norm $\|\cdot\|_\infty$) can be bounded as*

$$\log N\left(\epsilon M, C_M^N(\mathcal{X}), \|\cdot\|_\infty\right) \lesssim_{d,N,\mathsf{diam}(\mathcal{X})} \epsilon^{-d/N}, \; 0 < \epsilon \le 1,$$

*Proof of Lemma 2.* See Theorem 2.7.1 in [33]. □

We are now in position to prove Theorem 1.

*Proof of Theorem 1.* The proof applies Theorem 1.1 in [64] to the function class $\check{\mathcal{F}} = \check{\mathcal{F}}_{\sigma,d} = \{f * \varphi_\sigma : f \in \mathsf{Lip}_{1,0}\}$ to show that it is $P$-Donsker. We begin with noting that the function class $\check{\mathcal{F}}$ has envelope $\check{F}(x) := \check{F}_{\sigma,d}(x) := \|x\| + \sigma\sqrt{d}$. By assumption, $P\check{F}^2 < \infty$.

Next, for each $j$, consider the restriction of $\check{\mathcal{F}}$ to $I_j$, denoted as $\check{\mathcal{F}}_j = \{f\mathbb{1}_{I_j} : f \in \check{\mathcal{F}}\}$. To invoke [64, Theorem 1.1], we have to verify that each function class $\check{\mathcal{F}}_j$ is $P$-Donsker and to bound each $\mathbb{E}[\|\mathbb{G}_n\|_{\check{\mathcal{F}}_j}]$ where $\mathbb{G}_n := \sqrt{n}(P_n - P)$ and $\|\cdot\|_{\check{\mathcal{F}}_j} = \sup_{f \in \check{\mathcal{F}}_j} |\cdot|$. In view of Lemma 1, $\check{\mathcal{F}}_j$ can be regarded as a subset of $C_M^N(I_j)$ with $N = \lfloor d/2 \rfloor + 1$ and $M_j' = \left(\sup_{I_j} \|x\| + \sigma\sqrt{d}\right) \bigvee \sigma^{-\lfloor d/2 \rfloor} \sqrt{\lfloor d/2 \rfloor!}$. Thus, by Lemma 2, the $L^2(Q)$-metric entropy of $\check{\mathcal{F}}_j$ for any probability measure $Q$ on $\mathbb{R}^d$ can be bounded as

$$\log N\left(\epsilon M_j' Q(I_j)^{1/2}, \check{\mathcal{F}}_j, L^2(Q)\right) \lesssim_{d,K} \epsilon^{-d/(\lfloor d/2 \rfloor + 1)}.$$

The square root of the RHS is integrable (w.r.t. $\epsilon$) around 0, so that $\mathcal{F}_j$ is $P$-Donsker by Theorem 2.5.2 in [33], and by Theorem 2.14.1 in [33], we obtain

$$\mathbb{E}[\|\mathbb{G}_n\|_{\check{\mathcal{F}}_j}] \lesssim_{d,K} M_j' P(I_j)^{1/2} \lesssim_d \sigma^{-\lfloor d/2 \rfloor} M_j P(I_j)^{1/2}$$

with $M_j = \sup_{I_j} \|x\|$. By assumption, the RHS is summable over $j$.

By Theorem 1.1 in [64] we conclude that $\check{\mathcal{F}}$ is $P$-Donsker, which implies that there exists a tight version of $P$-Brownian bridge process $G_P$ in $\ell^\infty(\check{\mathcal{F}})$ such that $(\mathbb{G}_n f)_{f \in \check{F}}$ converges weakly in $\ell^\infty(\check{\mathcal{F}})$ to $G_P$. Finally, the continuous mapping theorem yields that

$$\sqrt{n}\mathsf{W}_1^{(\sigma)}(P_n, P) = \sup_{f \in \check{\mathcal{F}}} \mathbb{G}_n f \stackrel{d}{\to} \sup_{f \in \check{\mathcal{F}}} G_P(f) = \sup_{f \in \mathsf{Lip}_{1,0}} G_P^{(\sigma)}(f),$$

where $G_P^{(\sigma)}(f) := G_P(f * \varphi_\sigma)$. By construction, the Gaussian process $(G_P^{(\sigma)}(f))_{f \in \mathsf{Lip}_{1,0}}$ is tight in $\ell^\infty(\mathsf{Lip}_{1,0})$. The moment bound follows from summing up the moment bound for each $\check{\mathcal{F}}_j$. This completes the proof. □

### A.3  Proof of Corollary 1

We start with proving the following technical lemma.

**Lemma 3** (Distribution of $\boldsymbol{L_P^{(\sigma)}}$). *Assume the conditions of Theorem 1 and that $P$ is not a point mass. Then the distribution of $L_P^{(\sigma)}$ is absolutely continuous with respect to (w.r.t.) Lebesgue measure and its density is positive and continuous on $(0, \infty)$ except for at most countably many points.*

*Proof of Lemma 3.* From the proof of Theorem 1 and the fact that $\mathsf{Lip}_1$ is symmetric, we have $L_P^{(\sigma)} = \|G_P\|_{\check{\mathcal{F}}}$ with $\|\cdot\|_{\check{\mathcal{F}}} := \sup_{f \in \check{\mathcal{F}}} |\cdot|$. Since $G_P$ is a tight Gaussian process in $\ell^\infty(\check{\mathcal{F}})$,

$\check{\mathcal{F}}$ is totally bounded for the pseudometric $d_P(f, g) = \sqrt{\mathrm{Var}_P(f - g)}$, and $G_P$ is a Borel measurable map into the space of $d_P$-uniformly continuous functions $\mathcal{C}_u(\check{\mathcal{F}})$ equipped with the uniform norm $\|\cdot\|_{\check{\mathcal{F}}}$. Let $F$ denote the distribution function of $L_P^{(\sigma)}$, and define

$$r_0 := \inf\{r \geq 0 : F(r) > 0\}.$$

From [69, Theorem 11.1], $F$ is absolutely continuous on $(r_0, \infty)$, and there exists a countable set $\Delta \subset (r_0, \infty)$ such that $F'$ is positive and continuous on $(r_0, \infty) \setminus \Delta$. The theorem however does not exclude the possibility that $F$ has a jump at $r_0$, and we will verify that (i) $r_0 = 0$ and (ii) $F$ has no jump at $r = 0$, which lead to the conclusion. The former follows from p. 57 in [32]. The latter is trivial since

$$F(0) - F(0-) = \mathbb{P}\left(L_P^{(\sigma)} = 0\right) \leq \mathbb{P}\big(G_P(f) = 0\big),$$

for any $f \in \check{\mathcal{F}}$. Because $G_P$ is Gaussian we have $\mathbb{P}\big(G_P(f) = 0\big) = 0$ unless $f$ is constant $P$-a.s. $\qquad \square$

*Proof of Corollary 1.* From Theorem 3.6.2 in [33] applied to the function class $\check{\mathcal{F}}$, together with the continuous mapping theorem, we see that conditionally on $X_1, X_2, \ldots,$

$$\sqrt{n}\mathsf{W}_1^{(\sigma)}(P_n^B, P_n) = \sup_{f \in \check{\mathcal{F}}} \sqrt{n}(P_n^B - P_n)f \xrightarrow{d} L_P^{(\sigma)}$$

for almost every realization of $X_1, X_2, \ldots$ The desired conclusion follows from the fact that the distribution function of $L_P^{(\sigma)}$ is continuous (cf. Lemma 3) and Polya's theorem (cf. Lemma 2.11 in [70]). $\qquad \square$

## A.4 Proof of Corollary 3

Case (i) is Corollary 1 in [28]. Cases (ii) and (iii) follow from Theorems 4 and 2 in [71] and [72], respectively, applied to the function class $\check{\mathcal{F}}$ using the envelope function $\check{F}(x) = \|x\| + \sigma\sqrt{d}$. We omit the details for brevity. $\qquad \square$

# B Proofs for Section 4

## B.1 Preliminaries

The following technical lemmas will be needed.

**Lemma 4** (Continuity of $\mathsf{W}_1^{(\sigma)}$)**.** *The smooth Wasserstein distance $\mathsf{W}_1^{(\sigma)}$ is lower semicontinuous (l.s.c.) relative to the weak convergence on $\mathcal{P}(\mathbb{R}^d)$ and continuous in $\mathsf{W}_1$. Explicitly, (i) if $\mu_k \rightharpoonup \mu$ and $\nu_k \rightharpoonup \nu$, then*

$$\liminf_{k \to \infty} \mathsf{W}_1^{(\sigma)}(\mu_k, \nu_k) \geq \mathsf{W}_1^{(\sigma)}(\mu, \nu);$$

*and (ii) if $\mathsf{W}_1(\mu_k, \mu) \to 0$ and $\mathsf{W}_1(\nu_k, \nu) \to 0$, then*

$$\lim_{k \to \infty} \mathsf{W}_1^{(\sigma)}(\mu_k, \nu_k) = \mathsf{W}_1^{(\sigma)}(\mu, \nu). \tag{4}$$

*Proof.* Part (i). We first note that if $\mu_k \rightharpoonup \mu$, then $\mu_k * \mathcal{N}_\sigma \rightharpoonup \mu * \mathcal{N}_\sigma$. This follows from the facts that weak convergence is equivalent to pointwise convergence of characteristic functions, and the Gaussian measure has a nonvanishing characteristic function $\mathbb{E}_{X \sim \mathcal{N}_\sigma}[e^{it \cdot X}] = e^{-\sigma^2 \|t\|^2 / 2} \neq 0$ for all $t \in \mathbb{R}^d$. Now, if $\mu_k \rightharpoonup \mu$ and $\nu_k \rightharpoonup \nu$, then $\mu_k * \mathcal{N}_\sigma \rightharpoonup \mu * \mathcal{N}_\sigma$ and $\nu_k * \mathcal{N}_\sigma \rightharpoonup \nu * \mathcal{N}_\sigma$. From the lower semicontinuity of $\mathsf{W}_1$ relative to the weak convergence (cf. Remark 6.10 in [16]), we conclude that $\liminf_{k \to \infty} \mathsf{W}_1^{(\sigma)}(\mu_k, \nu_k) = \liminf_{k \to \infty} \mathsf{W}_1(\mu_k * \mathcal{N}_\sigma, \nu_k * \mathcal{N}_\sigma) \geq \mathsf{W}_1(\mu * \mathcal{N}_\sigma, \nu * \mathcal{N}_\sigma) = \mathsf{W}_1^{(\sigma)}(\mu, \nu)$.

Part (ii). Recall that $\mathsf{W}_1^{(\sigma)}$ generates the same topology as $\mathsf{W}_1$, i.e.,

$$\mathsf{W}_1^{(\sigma)}(\mu_k, \mu) \to 0 \iff \mathsf{W}_1(\mu_k, \mu) \to 0.$$

See Theorem 2 in [28]. So if $\mu_k \to \mu$ and $\nu_k \to \nu$ in $\mathsf{W}_1$, then $\mathsf{W}_1^{(\sigma)}(\mu_k, \mu) = \mathsf{W}_1(\mu_k * \mathcal{N}_\sigma, \mu * \mathcal{N}_\sigma) \to 0$ and $\mathsf{W}_1^{(\sigma)}(\nu_k, \nu) = \mathsf{W}_1(\nu_k * \mathcal{N}_\sigma, \nu * \mathcal{N}_\sigma) \to 0$. Thus, by Corollary 6.9 in [16], we have $\mathsf{W}_1^{(\sigma)}(\mu_k, \nu_k) = \mathsf{W}_1(\mu_k * \mathcal{N}_\sigma, \nu_k * \mathcal{N}_\sigma) \to \mathsf{W}_1(\mu_k * \mathcal{N}_\sigma, \nu_k * \mathcal{N}_\sigma) = \mathsf{W}_1^{(\sigma)}(\mu, \nu)$. $\qquad\square$

**Lemma 5** (Weierstrass criterion for the existence of minimizers). *Let $\mathcal{X}$ be a compact metric space, and let $f : \mathcal{X} \to \mathbb{R} \cup \{+\infty\}$ be l.s.c. (i.e., $\liminf_{x \to \overline{x}} f(x) \geq f(\overline{x})$ for any $\overline{x} \in \mathcal{X}$). Then, $\operatorname{argmin}_{x \in \mathcal{X}} f(x)$ is nonempty.*

*Proof.* See, e.g., p. 3 of [73]. $\qquad\square$

## B.2   Proof of Theorem 2

By Lemma 5, compactness of $\Theta$, and lower semicontinuity of the map $\theta \mapsto \mathsf{W}_1^{(\sigma)}(P_n(\omega), Q_\theta)$ (cf. Lemma 4), we see that $\operatorname{argmin}_{\theta \in \Theta} \mathsf{W}_1^{(\sigma)}(P_n(\omega), Q_\theta)$ is nonempty.

To prove the existence of a measurable estimator, we will apply Corollary 1 in [66]. Consider the empirical distribution as a function on $\mathcal{X}^{\mathbb{N}}$ with $\mathcal{X} = \mathbb{R}^d$, i.e., $\mathcal{X}^{\mathbb{N}} \ni x = (x_1, x_2, \dots) \mapsto P_n(x) = n^{-1} \sum_{i=1}^{n} \delta_{x_i}$. Observe that $\mathcal{X}^{\mathbb{N}}$ and $\mathbb{R}^{d_0}$ are both Polish, $\mathcal{D} := \mathcal{X}^{\mathbb{N}} \times \Theta$ is a Borel subset of the product metric space $\mathcal{X}^{\mathbb{N}} \times \mathbb{R}^{d_0}$, the map $\theta \mapsto \mathsf{W}_1^{(\sigma)}(P_n(x), Q_\theta)$ is l.s.c. by Lemma 4, and the set $\mathcal{D}_x = \{\theta \in \Theta : (x, \theta) \in \mathcal{D}\} \subset \mathbb{R}^{d_0}$ is $\sigma$-compact (as any subset in $\mathbb{R}^{d_0}$ is $\sigma$-compact). Thus, in view of Corollary 1 of [66], it suffices to verify that the map $(x, \theta) \mapsto \mathsf{W}_1^{(\sigma)}(P_n(x), Q_\theta)$ is jointly measurable.

To this end, we use the following fact: for a real function $\mathcal{Y} \times \mathcal{Z} \ni (y, z) \mapsto f(y, z) \in \mathbb{R}$ defined on the product of a separable metric space $\mathcal{Y}$ (endowed with the Borel $\sigma$-field) and a measurable space $\mathcal{Z}$, if $f(y, z)$ is continuous in $y$ and measurable in $z$, then $f$ is jointly measurable; see e.g. Lemma 4.51 in [74]. Equip $\mathcal{P}_1(\mathbb{R}^d)$ with the metric $\mathsf{W}_1$ and the associated Borel $\sigma$-field; the metric space $(\mathcal{P}_1(\mathbb{R}^d), \mathsf{W}_1)$ is separable [16, Theorem 6.16]. Then, since the map $\mathcal{X}^{\mathbb{N}} \ni x \mapsto P_n(x) \in \mathcal{P}_1(\mathbb{R}^d)$ is continuous (which is not difficult to verify), the map $\mathcal{X}^{\mathbb{N}} \times \Theta \ni (x, \theta) \mapsto (P_n(x), \theta) \in \mathcal{P}_1(\mathbb{R}^d) \times \Theta$ is continuous and thus measurable. Second, by Lemma 4, the function $\mathcal{P}_1(\mathbb{R}^d) \times \Theta \ni (\mu, \theta) \mapsto \mathsf{W}_1^{(\sigma)}(\mu, Q_\theta) \in [0, \infty)$ is continuous in $\mu$ and l.s.c. (and thus measurable) in $\theta$, from which we see that the map $(\mu, \theta) \mapsto \mathsf{W}_1^{(\sigma)}(\mu, Q_\theta)$ is jointly measurable. Conclude that the map $(x, \theta) \mapsto \mathsf{W}_1^{(\sigma)}(P_n(x), Q_\theta)$ is jointly measurable. $\qquad\square$

## B.3   Proof of Theorem 3

The proof relies on Theorem 7.33 in [67], and is reminiscent of that of Theorem B.1 in [37]; we present a simpler derivation under our assumption.[11] To apply Theorem 7.33 in [67], we extend the map $\theta \mapsto \mathsf{W}_1^{(\sigma)}(P_n, Q_\theta)$ to the entire Euclidean space $\mathbb{R}^{d_0}$ as

$$g_n(\theta) := \begin{cases} \mathsf{W}_1^{(\sigma)}(P_n, Q_\theta) & \text{if } \theta \in \Theta \\ +\infty & \text{if } \theta \in \mathbb{R}^{d_0} \setminus \Theta \end{cases}.$$

Likewise, define

$$g(\theta) := \begin{cases} \mathsf{W}_1^{(\sigma)}(P, Q_\theta) & \text{if } \theta \in \Theta \\ +\infty & \text{if } \theta \in \mathbb{R}^{d_0} \setminus \Theta \end{cases}.$$

The function $g_n$ is stochastic, $g_n(\theta) = g_n(\theta, \omega)$, but $g$ is non-stochastic. By construction, we see that $\operatorname{argmin}_{\theta \in \mathbb{R}^{d_0}} g_n(\theta) = \operatorname{argmin}_{\theta \in \Theta} \mathsf{W}_1^{(\sigma)}(P_n, Q_\theta)$ and $\operatorname{argmin}_{\theta \in \mathbb{R}^{d_0}} g(\theta) = \operatorname{argmin}_{\theta \in \Theta} \mathsf{W}_1^{(\sigma)}(P, Q_\theta)$. In addition, by Lemma 4, continuity of the map $\theta \mapsto Q_\theta$ relative to the weak topology, and closedness of the parameter space $\Theta$, we see that both $g_n$ and $g$ are l.s.c. (on $\mathbb{R}^{d_0}$). The main step of the proof is to show a.s. epi-convergence of $g_n$ to $g$. Recall the definition of epi-convergence (in fact, this is an equivalent characterization; see [67, Proposition 7.29]):

**Definition 1** (Epi-convergence). *For extended-real-valued functions $f_n$, $f$ on $\mathbb{R}^{d_0}$ with $f$ being l.s.c., we say that $f_n$ epi-converges to $f$ if the following two conditions hold:*

*(i)* $\liminf_{n\to\infty} \inf_{\theta\in\mathcal{K}} f_n(\theta) \geq \inf_{\theta\in\mathcal{K}} f(\theta)$ *for any compact set $\mathcal{K} \subset \mathbb{R}^{d_0}$; and*

*(ii)* $\limsup_{n\to\infty} \inf_{\theta\in\mathcal{U}} f_n(\theta) \leq \inf_{\theta\in\mathcal{U}} f(\theta)$ *for any open set $\mathcal{U} \subset \mathbb{R}^{d_0}$.*

We also need the concept of level-boundedness.

**Definition 2** (Level-boundedness). *For an extended-real-valued function $f$ on $\mathbb{R}^{d_0}$, we say that $f$ is level-bounded if for any $\alpha \in \mathbb{R}$, the set $\{\theta \in \mathbb{R}^{d_0} : f(\theta) \leq \alpha\}$ is bounded (possibly empty).*

We are now in position to prove Theorem 3.

*Proof of Theorem 3.* By boundedness of the parameter space $\Theta$, both $g_n$ and $g$ are level-bounded by construction as the (lower) level sets are included in $\Theta$. In addition, by assumption, both $g_n$ and $g$ are proper (an extended-real-valued function $f$ on $\mathbb{R}^{d_0}$ is proper if the set $\{\theta \in \mathbb{R}^{d_0} : f(\theta) < \infty\}$ is nonempty). In view of Theorem 7.33 in [67], it remains to prove that $g_n$ epi-converges to $g$ a.s. To verify property (i) in the definition of epi-convergence, recall that $P_n \to P$ in $\mathsf{W}_1$ (and hence in $\mathsf{W}_1^{(\sigma)}$) a.s. Pick any $\omega \in \Omega$ such that $P_n(\omega) \to P$ in $\mathsf{W}_1$. Pick any compact set $\mathcal{K} \subset \mathbb{R}^{d_0}$. Since $g_n(\cdot, \omega)$ is l.s.c., by Lemma 5, there exists $\theta_n(\omega) \in \mathcal{K}$ such that $g_n(\theta_n(\omega), \omega) = \inf_{\theta\in\mathcal{K}} g_n(\theta, \omega)$. Up to extraction of subsequences, we may assume $\theta_n(\omega) \to \theta^\star(\omega)$ for some $\theta^\star(\omega) \in \mathcal{K}$. If $\theta^\star(\omega) \notin \Theta$, then by closedness of $\Theta$, $\theta_n(\omega) \notin \Theta$ for all sufficiently large $n$. Thus, we have

$$\liminf_{n\to\infty} \inf_{\theta\in\mathcal{K}} g_n(\theta, \omega) = \liminf_{n\to\infty} g_n(\theta_n(\omega), \omega) = +\infty,$$

so that $\liminf_{n\to\infty} \inf_{\theta\in\mathcal{K}} g_n(\theta, \omega) \geq \inf_{\theta\in\mathcal{K}} g(\theta)$. Next, consider the case where $\theta^\star(\omega) \in \Theta$. In this case, $\theta_n(\omega) \in \Theta$ for all $n$ (otherwise, $+\infty = g_n(\theta_n(\omega), \omega) > g_n(\theta^\star(\omega), \omega)$, which contradicts the construction of $\theta_n(\omega)$). Thus, $g_n(\theta_n(\omega), \omega) = \mathsf{W}_1^{(\sigma)}(P_n(\omega), Q_{\theta_n(\omega)})$, so that

$$\begin{aligned}
\liminf_{n\to\infty} \inf_{\theta\in\mathcal{K}} g_n(\theta_n(\omega), \omega) &= \liminf_{n\to\infty} \mathsf{W}_1^{(\sigma)}(P_n(\omega), Q_{\theta_n(\omega)}) \\
&\overset{(a)}{\geq} \mathsf{W}_1^{(\sigma)}(P, Q_{\theta^\star(\omega)}) \\
&\geq \inf_{\theta\in\mathcal{K}} g(\theta),
\end{aligned} \tag{5}$$

where (a) follows from Lemma 4.

To verify property (ii) in the definition of epi-convergence, pick any open set $\mathcal{U} \subset \Theta$. It is enough to consider the case where $\mathcal{U} \cap \Theta \neq \varnothing$. Let $\{\theta'_n\}_{n=1}^\infty \subset \mathcal{U}$ be a sequence with $\lim_{n\to\infty} g(\theta'_n) = \inf_{\theta\in\mathcal{U}} g(\theta)$. Since $\inf_{\theta\in\mathcal{U}} g(\theta)$ is finite, we may assume that $\theta'_n \in \mathcal{U} \cap \Theta$ for all $n$. Thus, we have

$$\begin{aligned}
\limsup_{n\to\infty} \inf_{\theta\in\mathcal{U}} g_n(\theta, \omega) &\leq \limsup_{n\to\infty} g_n(\theta'_n, \omega) \\
&= \limsup_{n\to\infty} \mathsf{W}_1^{(\sigma)}(P_n(\omega), Q_{\theta'_n}) \\
&\leq \underbrace{\lim_{n\to\infty} \mathsf{W}_1^{(\sigma)}(P_n(\omega), P)}_{=0} + \underbrace{\lim_{n\to\infty} \mathsf{W}_1^{(\sigma)}(P, Q_{\theta'_n})}_{=\inf_{\theta\in\mathcal{U}} g(\theta)} \\
&= \inf_{\theta\in\mathcal{U}} g(\theta).
\end{aligned} \tag{6}$$

Conclude that $g_n$ epi-converges to $g$ a.s. This completes the proof. $\qquad\square$

## B.4 Proof of Theorem 4

Recall that $P = Q_{\theta^\star}$. Condition (ii) implies that $\operatorname{argmin}_{\theta\in\Theta} \mathsf{W}_1^{(\sigma)}(P, Q_\theta) = \{\theta^\star\}$. Hence, by Theorem 3, for any neighborhood $N$ of $\theta^\star$,

$$\inf_{\theta\in\Theta} \mathsf{W}_1^{(\sigma)}(P_n, Q_\theta) = \inf_{\theta\in N} \mathsf{W}_1^{(\sigma)}(P_n, Q_\theta)$$

with probability approaching one.

Define $R_\theta^{(\sigma)} := Q_\theta^{(\sigma)} - P^{(\sigma)} - \langle \theta - \theta^\star, D^{(\sigma)} \rangle \in \ell^\infty(\mathsf{Lip}_{1,0})$, and choose $N_1$ as a neighborhood of $\theta^\star$ such that

$$\left\| \langle \theta - \theta^\star, D^{(\sigma)} \rangle \right\|_{\mathsf{Lip}_{1,0}} - \left\| R_\theta^{(\sigma)} \right\|_{\mathsf{Lip}_{1,0}} \geq \frac{1}{2} C, \quad \forall \theta \in N_1, \tag{7}$$

for some constant $C > 0$. Such $N_1$ exists since conditions (iii) and (iv) ensure the existence of an increasing function $\eta(\delta) = o(1)$ (as $\delta \to 0$) and a constant $C > 0$ such that $\left\| R^{(\sigma)}(\theta) \right\|_{\mathsf{Lip}_{1,0}} \leq \|\theta - \theta^\star\| \eta(\|\theta - \theta^\star\|)$ and $\left\| \langle t, D^{(\sigma)} \rangle \right\|_{\mathsf{Lip}_{1,0}} \geq C\|t\|$ for all $t \in \mathbb{R}^{d_0}$.

For any $\theta \in N_1$, the triangle inequality and (7) imply that

$$\mathsf{W}_1^{(\sigma)}(P_n, Q_\theta) \geq \frac{C}{2} \|\theta - \theta^\star\| - \mathsf{W}_1^{(\sigma)}(P_n, P). \tag{8}$$

For $\xi_n := \frac{4\sqrt{n}}{C} \mathsf{W}_1^{(\sigma)}(P_n, P)$, consider the (random) set $N_2 := \{\theta \in \Theta : \sqrt{n}\|\theta - \theta^\star\| \leq \xi_n\}$. Note that $\xi_n$ is of order $O_\mathbb{P}(1)$ by Theorem 1. By the definition of $\xi_n$, $\inf_{\theta \in N_1} \mathsf{W}_1^{(\sigma)}(P_n, Q_\theta)$ is unchanged if $N_1$ is replaced with $N_1 \cap N_2$; indeed, if $\theta \in N_2^c$, then $\mathsf{W}_1^{(\sigma)}(P_n, Q_\theta) > \frac{C}{2} \frac{\xi_n}{\sqrt{n}} - \mathsf{W}_1^{(\sigma)}(P_n, P) = \mathsf{W}_1^{(\sigma)}(P_n, P)$, so that $\inf_{\theta \in N_2^c} \mathsf{W}_1^{(\sigma)}(P_n, P) > \mathsf{W}_1^{(\sigma)}(P_n, P) \geq \inf_{\theta \in N_1} \mathsf{W}_1^{(\sigma)}(P_n, Q_\theta)$.

Reparametrizing $t := \sqrt{n}(\theta - \theta^\star)$ and setting $T_n := \{t \in \mathbb{R}^{d_0} : \|t\| \leq \xi_n, \ \theta^\star + t/\sqrt{n} \in \Theta\}$, we have the following approximation

$$\sup_{t \in T_n} \left| \sqrt{n} \underbrace{\left\| P_n^{(\sigma)} - Q_{\theta^\star + t/\sqrt{n}}^{(\sigma)} \right\|_{\mathsf{Lip}_{1,0}}}_{=\mathsf{W}_1^{(\sigma)}(P_n, Q_{\theta^\star + t/\sqrt{n}})} - \left\| \underbrace{\sqrt{n}\big(P_n^{(\sigma)} - P^{(\sigma)}\big)}_{=\mathbb{G}_n^{(\sigma)}} - \langle t, D^{(\sigma)} \rangle \right\|_{\mathsf{Lip}_{1,0}} \right|$$
$$\leq \sup_{t \in T_n} \sqrt{n} \left\| R_{\theta^\star + t/\sqrt{n}}^{(\sigma)} \right\|_{\mathsf{Lip}_{1,0}} \tag{9}$$
$$\leq \xi_n \eta(\xi_n/\sqrt{n})$$
$$= o_\mathbb{P}(1).$$

Observe that any minimizer $t^\star \in \mathbb{R}^{d_0}$ of the function $h_n(t) := \left\| \mathbb{G}_n^{(\sigma)} - \langle t, D^{(\sigma)} \rangle \right\|_{\mathsf{Lip}_{1,0}}$ satisfies $\|t^\star\| \leq \xi_n$; indeed if $\|t^\star\| > \xi_n$, then $h_n(t^\star) \geq C\|t^\star\| - \|\mathbb{G}_n^{(\sigma)}\|_{\mathsf{Lip}_{1,0}} = C\|t^\star\| - \sqrt{n} \mathsf{W}_1^{(\sigma)}(P_n, P) = 3\sqrt{n} \mathsf{W}_1^{(\sigma)}(P_n, P) = 3h_n(0)$, which contradicts the assumption that $t^\star$ is a minimizer of $h_n(t)$. Since by assumption $\theta^\star \in \mathrm{int}(\Theta)$, the set of minimizers of $h_n$ lies inside $T_n$. Conclude that

$$\inf_{\theta \in \Theta} \sqrt{n} \mathsf{W}_1^{(\sigma)}(P_n, Q_\theta) = \inf_{t \in \mathbb{R}^{d_0}} \left\| \mathbb{G}_n^{(\sigma)} - \langle t, D^{(\sigma)} \rangle \right\|_{\mathsf{Lip}_{1,0}} + o_\mathbb{P}(1). \tag{10}$$

Now, from the proof of Theorem 1 and the fact that the map $G \mapsto (G(f * \varphi_\sigma))_{f \in \mathsf{Lip}_{1,0}}$ is continuous (indeed, isometric) from $\ell^\infty(\check{\mathcal{F}})$ into $\ell^\infty(\mathsf{Lip}_{1,0})$, we see that $(\mathbb{G}_n^{(\sigma)} f)_{f \in \mathsf{Lip}_{1,0}} \to G_P^{(\sigma)}$ weakly in $\ell^\infty(\mathsf{Lip}_{1,0})$

Applying the continuous mapping theorem to $L \mapsto \inf_{t \in \mathbb{R}^{d_0}} \left\| L - \langle t, D^{(\sigma)} \rangle \right\|_{\mathsf{Lip}_{1,0}}$ and using the approximation (10), we obtain the conclusion of the theorem. $\qquad\square$

## B.5  Proof of Corollary 2

The proof relies on the following result on weak convergence of argmin solutions of convex stochastic functions. The following lemma is a simple modification of Theorem 1 in [75]. Similar techniques can be found in [76] and [77].

**Lemma 6.** *Let $H_n(t)$ and $H(t)$ be convex stochastic functions on $\mathbb{R}^{d_0}$. Suppose that (i) $\operatorname{argmin}_{t \in \mathbb{R}^{d_0}} H(t)$ is unique a.s., and (ii) for any finite set of points $t_1, \ldots, t_k \in \mathbb{R}^{d_0}$, we have $(H_n(t_1), \ldots, H_n(t_k)) \xrightarrow{d} (H(t_1), \ldots, H(t_k))$. Then, for any sequence $\{\widehat{t}_n\}_{n \in \mathbb{N}}$ such that $H_n(\widehat{t}_n) \leq \inf_{t \in \mathbb{R}^{d_0}} H_n(t) + o_\mathbb{P}(1)$, we have $\widehat{t}_n \xrightarrow{d} \operatorname{argmin}_{t \in \mathbb{R}^{d_0}} H(t)$.*

(a) Mean-Variance estimation

(b) Mean estimation for 2-mode Gaussian mixture

Figure 4: One-dimensional limiting distributions for: (a) the mean and variance of an MSWE-based generative model fitted to $P = \mathcal{N}(\mu_\star, \sigma_\star^2)$, with $\mu_\star = 0$ and $\sigma_\star = 1$; and (b) the two mean parameters of the mixture $P = 0.5\mathcal{N}(\mu_1, 1) + 0.5\mathcal{N}(\mu_2, 1)$, for $\mu_1 = 0$ and $\mu_2 = 1$. Also shown on a log-log scale (with error bars) is the SWD convergence as a function of $n$.

*Proof of Corollary 2.* By Theorem 3, $\widehat{\theta}_n \to \theta^\star$ in probability. From equation (8) and the definition of $\widehat{\theta}_n$, we see that, with probability approaching one,

$$\underbrace{\inf_{\theta \in \Theta} \sqrt{n}\mathsf{W}_1^{(\sigma)}(P_n, Q_\theta)}_{=O_\mathbb{P}(1)} + o_\mathbb{P}(1) \geq \sqrt{n}\mathsf{W}_1^{(\sigma)}(P_n, Q_{\widehat{\theta}_n}) \geq \frac{C}{2}\sqrt{n}\|\widehat{\theta}_n - \theta^\star\| - \underbrace{\sqrt{n}\mathsf{W}_1^{(\sigma)}(P_n, P)}_{=O_\mathbb{P}(1)},$$

which implies that $\sqrt{n}\|\widehat{\theta}_n - \theta^\star\| = O_\mathbb{P}(1)$. Let $H_n(t) := \left\|\mathbb{G}_n^{(\sigma)} - \left\langle t, D^{(\sigma)}\right\rangle\right\|_{\mathsf{Lip}_{1,0}}$ and $H(t) := \left\|\mathbb{G}_P^{(\sigma)} - \left\langle t, D^{(\sigma)}\right\rangle\right\|_{\mathsf{Lip}_{1,0}}$. Both $H_n(t)$ and $H(t)$ are convex in $t$. Then, from equation (9), for $\widehat{t}_n := \sqrt{n}(\widehat{\theta}_n - \theta^\star) = O_\mathbb{P}(1)$, we have

$$\sqrt{n}\mathsf{W}_1^{(\sigma)}(P_n, Q_{\widehat{\theta}_n}) = H_n(\widehat{t}_n) + o_\mathbb{P}(1).$$

Combining the result (10) and the definition of $\widehat{\theta}_n$, we see that $H_n(\widehat{t}_n) \leq \inf_{t \in \mathbb{R}^{d_0}} H_n(t) + o_\mathbb{P}(1)$. Since $\mathbb{G}_n^{(\sigma)}$ converges weakly to $G_P^{(\sigma)}$ in $\ell^\infty(\mathsf{Lip}_{1,0})$, by the continuous mapping theorem, we have $(H_n(t_1), \ldots, H_n(t_k)) \xrightarrow{d} (H(t_1), \ldots, H(t_k))$ for any finite number of points $t_1, \ldots, t_k \in \mathbb{R}^{d_0}$. By assumption, $\operatorname{argmin}_{t \in \mathbb{R}^{d_0}} H(t)$ is unique a.s. Hence, by Lemma 6, we conclude that $\widehat{t}_n \xrightarrow{d} \operatorname{argmin}_{t \in \mathbb{R}^{d_0}} H(t)$. $\square$

**Remark 6** (Alternative proofs). *Corollary 2 alternatively follows from the proof of Theorem 4 combined with the argument given at the end of p. 63 in [2] (plus minor modifications), or the result of Theorem 5 combined with the argument given at the end of p. 67 in [2]. The proof provided above is differs from both these arguments and is more direct.*

## C Additional Experiments

Figure 4 shows tne-dimensional limiting distributions for: (a) the mean and variance of an MSWE-based generative model fitted to $P = \mathcal{N}(\mu_\star, \sigma_\star^2)$, with $\mu_\star = 0$ and $\sigma_\star = 1$; and (b) the two mean parameters of the mixture $P = 0.5\mathcal{N}(\mu_1, 1) + 0.5\mathcal{N}(\mu_2, 1)$, for $\mu_1 = 0$ and $\mu_2 = 1$ (repeated from the main text). Also shown on a log-log scale (with 1-sigma error bars) is the SWD convergence as a function of $n$.

## Footnotes

[11]Theorem B.1 in [37] applies Theorem 7.31 in [67]. To that end, one has to extend the maps $\theta \mapsto \mathcal{W}_p(\widehat{\mu}_n, \mu_\theta)$ and $\theta \mapsto \mathcal{W}_p(\mu_\star, \mu_\theta)$ to the entire Euclidean space $\mathbb{R}^{d_\theta}$. The extension was not mentioned in the proof of [37, Theorem B.1], although this missing step does not affect their final result.