[Reviews · NeurIPS 2020]

Review 1

Summary and Contributions: ***Update after author feedback*** My initial impression stands. I think this is a very good paper relatively to other submissions I have reviewed currently and in the past for ML conferences. In my opinion, the question tackled by the authors is hard, relevant, and the theoretical results are not trivial at all. I agree with the author rebuttal that the results are of statistical, rather than computational nature. However, computational limitations should be put more upfront, as they are mentioned only briefly. On the other hand, my main concern remains: the discretization error of SWD still depends exponentially on the dimension (ref [26]) but in this manuscript they are conveniently hidden in the approx less than or equal notation, which I think is misleading and should be remarked. Rebuttal has clarified my concerns about figure 1, and I agree it would be good to expand on its description. ***Original Review*** The paper derives limit distributions for the smoothed wasserstein distance between a probability measure and its empirical counterpart, as the number of samples tend to infinity. The smoothed Wasserstein distance is the usual Wasserstein distance between two measure but convolved with gaussian noise. Such results were only known before in one dimension, or for measures supported on a countable set. Measurability, consistency and limiting distributions of the minimum smooth wasserstein distance are also derived.

Strengths: The paper is highly significant as it develops further the theory around a recently proposed smooth wasserstein distance that attempts to overcome the curse of dimensionality of the original W1. The work is highly relevant to the community as optimal transport has been used in many machine learning applications in recent years. The clarity of presentation and mathematical rigour is another strength of this paper, as it states the results in a precise and concise manner, while keeping a simple notation. The quality of the provided proofs and the general structure and style of the paper show that the authors have a deep understanding of the topic.

Weaknesses: The main weakness I find is that Figure 1 is hard to understand visually, as the point clouds all seem to be pretty similar, I have a hard time understanding if there should be a difference between the clouds corresponding to different colors. Perhaps the authors should elaborate a bit more on the main takeaway from this figure, and what should we expect from the point clouds corresponding to different colors.

Correctness: I have checked the high-level points of the proofs for soundness and they seem correct, however I would say such results merit further reviewing that what a conference review cycle allows. Claims and arguments are clear and well written. The results are built with a mix of new and well-known techniques, and rely also on well stablished results in statistics and probability. The proposed experiments illustrate well the theory. However I was somewhat confused by Figure 1 (see weaknesses section).

Clarity: The paper is really well written, it is easy to follow and the main claims and statements of the theorems are precise. The level of rigour is high while preserving readability.

Relation to Prior Work: The main differences with previous work are clearly stated and authors provide enough references to relevant material.

Reproducibility: Yes

Additional Feedback: 1. Integral probability metrics may attain the parametric rate of convergence n^(-1/2) see for example https://papers.nips.cc/paper/6483-minimax-estimation-of-maximum-mean-discrepancy-with-radial-kernels.pdf so please check your claim in line 34. Also it is not so clear to mention wass. distance and IPMs as different things (lines 33,34) because W1 is a particular case of an IPM. 2. Line 45 is a bit misleading it might be better to recall that the rate of W1^sigma still depends exponentially on the dimension 3. Line 56: allows enables


Review 2

Summary and Contributions: After reading the rebuttal, I slightly increased my grade. I think the authors only partially addressed the issue I raised. In particular, the statement "exponential in d runtime (though no major issues arose in practice)" is quite problematic and suggests that the authors are not taking seriously the numerical evaluation of the theory. == This paper proposes a theoretical study of Gaussian smoothing of optimal transport. In particular it performs a sort of central limit theorem analysis, therefore extending some previously known results in the case of (un-regularized) Wasserstein distances.

Strengths: It studies an important problem, is well written and the mathematical treatment is rigorous. I would be quite supportive of its acceptance in case the issues raised below are addressed. I think this should include a re-writing of some parts, and this could be explained in the rebuttal.

Weaknesses: I think the paper vastly oversells the Gaussian smoothing technic, to the point that it deserves its interesting mathematical results. In short, I think the paper does not provide a fair overview of the current bottlenecks of using OT in high dimension, mostly because: 1/ up to know estimating Gaussian smoothing distance seems intractable, 2/ there is no clear statement about why is W^sigma an interesting quantity to minimize. Either it is because one is interested in approximation OT (but this it not the problem under study) or it is because it carries interesting properties for a non-vanishing sigma, but this is not really discussed. Quoting the authors "W^sigma enjoys the rich structure of Wasserstein distances" but I am unsure what it means. I would love to be wrong on both issues (and I might be wrong), but at very least I find that the paper does not really answer these key issues and hides the most important and interesting questions of OT for ML. In the current state of the paper (in particular given the numerical schemes proposed, using re-sampling and GAN-like methods), I think the statement "posing the SWD as a potent tool for learning and inference in high dimensions" is not correct.

Correctness: From a theoretical perspective the paper is correct. The numerical part however is not really satisfying (even as a proof of concept to validate the theory) as explained below.

Clarity: Yes the paper is well written.

Relation to Prior Work: As explained bellow, a lot of relevant literature is missing. -- Bibliography -- [1] Jonathan Weed and Quentin Berthet. Estimation of smooth densities in Wasserstein distance. In Conference on Learning Theory, pages 3118–3119, 2019. [2] Gonzalo Mena and Jonathan Niles-Weed. Statistical bounds for entropic optimal transport: sample complexity and the central limit theorem. In Advances in Neural Information Processing Systems, pages 4543–4553, 2019. [3] Aude Genevay, Lénaïc Chizat, Francis Bach, Marco Cuturi, and Gabriel Peyré. Sample complexity of Sinkhorn divergences. In The 22nd International Conference on Artificial Intelligence and Statistics, pages 1574–1583, 2019. [4] Central limit theorems for entropy-regularized optimal transport on finite spaces and statistical applications Jérémie Bigot, Elsa Cazelles, Nicolas Papadakis [5] M. Cuturi, Sinkhorn Distances: Lightspeed Computation of Optimal Transportation Distances

Reproducibility: Yes

Additional Feedback: The list of remarks and questions I have: * The current standard for regularization of OT is entropic regularization of the plan (papers of Cuturi [5], and also sample complexity results [3,4]). This paper seems to mostly ignore this literature, which is quite weird, given the fact that the goals are (almost) the same. * There are known results on the asymptotic distribution of entropic regularized transport, see for instance [4] (there might be other references, including studies of parameter estimation). Given the fact that entropic regularization can (should) be viewed as a "cheap proxy" for Gaussian smoothing, a proper and detailed comparison seems in order. * The biggest issue of the paper (unless I am missing something) is the actual computation of W^sigma. The authors seem to be using a re-sampling scheme "Sampling from P_n and N(sigma)_x0000_ and adding the obtained values produces samples from P_n * N(sigma)". But the potential problem is that to cope with the CoD, this might requires a number of samples exponential in the dimension. * Gaussian smoothing is very similar to technic as the one proposed [1] to fight the curse of dimensionality for smooth densities (they use a more involved wavelet method, but for uniformly smooth densities this would have the same properties). They do not propose efficient algorithms, and use a re-sampling scheme and a discrete plugin estimator, which I have the impression is the same at what is used here. Some comments on this type of approaches seem in order. * The use of a GAN-type method is also problematic and not covered by the theory I believe. * This is in sharp contrast with Sinkhorn's approach, which seems to offer similar 1/sqrt(n) sample complexity guarantees, could be amenable to a similar analysis of its asymptotic distributions (see the reference above), but at a cheaper O(n^2) computational price, independent of the dimension. This should at least be discussed. * The role of sigma>0 is not really explained. For instance, the dependency on sigma of the constants involved is not discussed. * The extreme case of Sinkhorn regularization is MMD. It is also the asymptotic of large sigma smoothing. MMD does not suffer either from the curse of dimensionality. So what would be the type of tradeoffs in term of parameter selection vs sample size and smoothness of the distributions governing the use of Gaussian smoothing, Sinkhorn smoothing and MMD ?


Review 3

Summary and Contributions: This paper deals with minimum distance estimation, i.e., estimation of a parameter $\theta$ by minimizing (with respect to it) a statistical distance between the observed empirical distribution and the target one (which depends on $\theta$), when the statistical distance at stake is the smooth 1-Wasserstein distance. The main contribution of this paper is to establish asymptotic results for the smooth 1-Wasserstein estimator and related quantities, when the underlying distribution is sub-Gaussian. == Update == I would like to thank the authors for their response. They plan to modify the manuscript based on my main remarks and their feedback is satisfactory in this regard. My opinion on this paper remains very positive.

Strengths: Wasserstein distances (and others) suffer from the curse of dimensionality: the rate of convergence of the empirical distribution is of order $n^{-1/d}$, which is obviously less and less good as the dimension $d$ increases. In contrast, the paper states that the rate of convergence under the smooth Wasserstein distance is $n^{-1/2}$ whatever the dimension when the distribution is sub-Gaussian. This main result is to me of great interest for the community and I felt particularly enthusiastic when reading the paper.

Weaknesses: From my point of view, the main limitation of this work lies in the choice of the smoothing parameter $\sigma$, which is unfortunately never discussed. From both the theoretical results (Theorem 1) and the numerical simulation study, one may think that the best choice is to take it very large, but in that case, the smooth Wasserstein distance can be far from the usual one. The choice of a statistical distance can not be done only on its asymptotic properties, but also on what it measures on the data. What is the opinion of the authors on this question? Perhaps it would have been a good idea to compare generalization results from Wasserstein and smooth Wasserstein distances.

Correctness: To the best of my understanding, the results presented in this paper are correct.

Clarity: The paper is pretty easy to read although technical. I have the following two minor remarks: - The paper is presented with a strong emphasis on generative modeling, but the theoretical and numerical results it contains, although very interesting, concern classical statistical problems (parametric estimation). Despite the paragraph "Generalization discussion" (l.248), I think that the content of this manuscript could benefit from another presentation. - In Theorem 1, the order of the dependency on $\sigma$ is easy to read, but this is not the case in other results. I understand that this can be very tricky and technical, especially because of the definition of the limit process, but perhaps the authors could help the reader on this point. Typos l.56: allows enables l.95: sharp rate are l.138: one many then obtain l.279: the distribution [...] converge

Relation to Prior Work: Sliced Wasserstein distance has similar properties to the smooth Wasserstein distance investigated in this paper since both do not suffer from the curse of dimensionality (see Theorems 5 and 6 in Nadjahi et al., NeurIPS 2019). I would have liked the positioning in relation to this work to be more detailed than the few lines written in the introduction. Perhaps it goes beyond the scope of the paper, but a natural question is: which distance to choose since both do not suffer from the dimensionality problem? It can be from a theoretical or a very practical point of view.

Reproducibility: Yes

Additional Feedback: See my questions and comments in the previous sections: weaknesses, clarity, and relation to prior work.


Review 4

Summary and Contributions: +++++++++ Update after reading the author's response +++++++++ I thank the authors for their response. I however agree with reviewer #3 in that the claim that the rate of convergence is independent of the sample dimension should be tempered in view of the computational reality at stake with the use of the metric. In particular, as I mentioned in my review, this needs to be addressed via a more thorough exposition of the GAN experiment. +++++++++++++++++++++++++++++++++++++++++++++ The paper establishes new, nice and important theoretical properties of the smooth Wasserstein distance (SWD) as a statistical distance for learning generative models. In particular, the convergence with rate independent of the sample dimension (of order square root of the sample size) is proved for the parameter of the distribution minimizing the SWD to the empirical distribution of the sample.

Strengths: The paper is well written and provides proofs or references for all stated results. It provides new results which look very promising for the machine learning community by establishing statistical properties of a statistical distance which are independent of the data dimension. These results are very likely to draw attention to the smooth Wassertein distance in the ML community as this statistical distance is showed to be quite unique in its robustess to the curse of dimensionality. The authors also provide convincing empirical evidence of their result.

Weaknesses: While the paper is well written for mathematical standards, the importance of the work for ML applications may be difficult to grasp for readers with a more empirical background. In particular, the presentation of the last experiment using a parametric family of distributions given by a generative network can be improved.

Correctness: I do not have the expertise to check the proof in the amount of time provided, but I am confident that an expert in the field should have no problem doing this job given the clarity of the authors' exposition. The empirical methodology is correct, even though more details are necessary for the last experiment.

Clarity: Yes, as mentioned already. However more effort can be made to meet the wider audience interested in the results. Note the following typos: 56 choose between allows or enables 95 rateS 511 have TO verify

Relation to Prior Work: This is done with much details.

Reproducibility: Yes

Additional Feedback:

[Author Response · NeurIPS 2020]

We thank the reviewers for the positive feedback and valuable comments. Our response follows.

**1) Computing smooth Wasserstein distance (SWD) [R3]:** This is a great and important point. We will add a discus-
sion about computation to the revision. As R3 mentions, the placeholder MC algorithm we used, in theory, has an
exponential in $d$ runtime (though no major issues arose in practice). Our submission studies statistical aspects of SWD,
but we are actively working on efficient algorithms that compute it. To avoid sampling the kernel, we developed a
method to compute $\mathsf{W}_1^{(\sigma)}(P_n,Q_m)=\sup_{f\in\mathsf{Lip}_1}P_n(f*\varphi_\sigma)-Q_m(f*\varphi_\sigma)$ under NN parameterization of $f$, with the high-
dimensional convolution $f*\varphi_\sigma$ implemented in closed form. Accuracy and complexity of the alg. are being explored.

**2) Choice of SD [R3,R4]:** We will revise the presentation to thoroughly motivate SWD for inference. Briefly, SWD
inherits much of its structure from classic $\mathsf{W}_1$ [27], while enjoying faster empirical convergence. [27] shows: (i) SWD
metrizes the same topology as $\mathsf{W}_1$; (ii) it $\Gamma$-converges to $\mathsf{W}_1$ as $\sigma\to 0$ (with convergence of optimizers); (iii) it is stable
in $\sigma$, $|\mathsf{W}_1^{(\sigma)}(P,Q)-\mathsf{W}_1^{(\tau)}(P,Q)|\leq 2\sqrt{d|\sigma^2-\tau^2|}$; (iv) $\mathsf{W}_1^{(\sigma)}$ is continuous and nonincreasing in $\sigma$. These suggest that
SWD preserves the compatibility of $\mathsf{W}_1$ for inference, while having superior statistical properties. Re generalization,
Supp. A.1 states concentration inequalities for SWD, which imply generalization bounds for SMDE in the sense of [14]
(lines 248-255). SWD achieves fast $n^{-1/2}$ generalization, as opposed to the $n^{-1/d}$ generalization gap of $\mathsf{W}_1$.

**3) Effect/choice of $\sigma$ [R2,R3,R4]:** Item 2 lists properties of $\mathsf{W}_1^{(\sigma)}$ as a function of $\sigma$. From a statistical standpoint, the
expected empirical SWD scales as $\sigma^{-d/2}n^{-1/2}$ (see [26] and Thm. 1 herein). We are currently working on relaxing the
prefactor under the manifold hypothesis, i.e., showing that it depends only on the manifold's (rather than the ambient)
dimension. R4 is correct that characterizing the dependence on $\sigma$ in limit distribution results is nontrivial, and we
will comment on that in the revision, as suggested. In practice, having an excessively large $\sigma$ may result in generative
models that are slow to learn small-scale structures in the target distribution (though in theory they should be recovered
eventually). That said, SWD is remarkably stable over moderate changes in $\sigma$ and is thus not hard to tune.

**4) Sliced $\mathsf{W}_1$ ($\mathsf{SW}_1$) vs. $\mathsf{W}_1^{(\sigma)}$[R4]:** We will gladly expand the $\mathsf{SW}_1$ discussion as follows. $\mathsf{SW}_1$ and SWD are different
approaches for alleviating the curse of dimensionality (CoD) of $\mathsf{W}_1$. $\mathsf{SW}_1$ uses 1D projections, while SWD levels out
local irregularities in the distributions by convolving with $\mathcal{N}_\sigma$. $\mathsf{SW}_1$ and SWD share many similarities, e.g., both are
metrics and (topologically) equivalent to $\mathsf{W}_1$. $\mathsf{SW}_1$ is easily computed using the 1D formula, while computational
aspects of SWD are still in the works (Item 1). SWD might be preferable when comparing to regular $\mathsf{W}_1$, as it is within an
additive $2\sigma\sqrt{d}$ gap from $\mathsf{W}_1$ [27] (see Item 2). Comparison results for $\mathsf{SW}_1$ seem weaker, assuming compact support and
involving implicit dimension-dependent constants (cf., e.g., Lem. 5.1.4. in http://cvgmt.sns.it/paper/2341/).

**5) Entropic OT (EOT) vs. $\mathsf{W}_1^{(\sigma)}$[R3]:** Generally, we do not view SWD and EOT as competing techniques. It is
beneficial for the community to have several methods for dealing with the CoD, especially since tradeoffs often emerge.
We agree that a thorough literature review of EOT is appropriate, and will cover the following in the revision: (i) EOT
is not a metric, while SWD retains the metric structure of $\mathsf{W}_1$; (ii) $n^{-1/2}$ rate for EOT is proven only for smooth costs
(excluding $\mathsf{W}_1$) with compactly supported distributions (arXiv:1810.02733) or squared cost with subgassian distributions
(arXiv:1905.11882), while for SWD the $n^{-1/2}$ rate holds under mild polynomial moment conditions; (iii) EOT CLT
(arXiv:1905.11882) is similar to (arXiv:1705.01299) but markedly different from ours. Notably, they derive the result
for two-sample populations with the unknown centering constants $\mathbb{E}[S_\epsilon(P_n,Q)]$ or $\mathbb{E}[S_\epsilon(P_n,Q_m)]$, which differ from
$S_\epsilon(P,Q)$. (iv) A major virtue of EOT lies in fast algs., and matching those for SWD is a central goal going forward.

**6) $\mathsf{W}_1^{(\sigma)}$vs. MMD and $\sigma\to\infty$ limit [R2,R3]:** MMDs do not suffer from the CoD, due to low complexity of RKHS
function class. A main motivator for studying SWD is to alleviate the Wasserstein CoD, while preserving the metric
structure. In light of Item 2, we are thus interested in $\sigma<\infty$. Also, [Cor. 2.4, arXiv:2005.00738] shows that
$\lim_{\sigma\to\infty}\mathsf{W}_1^{(\sigma)}(P,Q)=|\mathbb{E}_P[X]-\mathbb{E}_Q[Y]|$, which is not informative as a discrepancy measure between distributions.

**7) Weed-Berthet [R3]:** Their focus is different from ours: they study density estimation under $\mathsf{W}_p$ and do not deal
with limit distributions or MDE. Re computation, please see Item 1. Still, we are glad to cite this paper in the revision.

**8) Applying analysis to EOT [R3]:** Our proof technique to derive the asymptotic distribution relies on expressing
SWD *exactly* as the supremum of an empirical process indexed by smoothed $\mathsf{Lip}_1$ functions. As EOT cannot be written
in this manner (for some popper function class), our proof technique does not directly extend to EOT.

**9) Fig. 1 [R2]:** The purpose of the point clouds is to show that the point clouds closely overlap. This implies that indeed
a limiting distribution is emerging. We will add additional description to make this much more clear in the revision.

**10) GAN method [R3], parametric experiments [R4], and appeal to ML audience [R6]:** When referring to GAN-
type methods, does R3 mean the generator's parameterization or the adversarial training? The former is covered by the
MDE theory by viewing the weights/biases of the generator NN as $\theta$ (adversarial training aspects are indeed beyond the
scope of this work). Our last experiment in fact implements a SWD GAN: while the data comes from a parametric
distribution (GMM with $\exp(d)$ modes) we match it with a (generator) NN transformation of a latent variable. Per R6's
suggestion, we will revise the presentation of this experiment to better communicate it to ML audiences. In general we
will put more emphasis on NN-based ML applications, in addition to our focus on parametric setups.

[Meta-Review · NeurIPS 2020]

The reviewers agree that this is a good paper that deserves acceptance. The contributions are useful from a statistical point of view. They also agree that the computational limitations should be put more upfront: the idea of Gaussian smoothing has a limited interest for the neurips community unless one has an efficient algorithm to solve optimal transport between the smoothed densities, which is not the case yet (any method based purely on discretizations, as proposed here, inevitably suffers from the curse of dimensionality). The authors mention in the rebuttal that an idea is to parameterize the dual variable with a neural network, but this leads to an object that is very different from SWD since neural networks have inductive biases. For these reasons, I recommend accept (poster).